# TEMPORAL DIFFERENCE UNCERTAINTIES AS A SIGNAL FOR EXPLORATION

## ABSTRACT

An effective approach to exploration in reinforcement learning is to rely on an agent's uncertainty over the optimal policy, which can yield near-optimal exploration strategies in tabular settings. However, in non-tabular settings that involve function approximators, obtaining accurate uncertainty estimates is almost as challenging as the exploration problem itself. In this paper, we highlight that value estimates are easily biased and temporally inconsistent. In light of this, we propose a novel method for estimating uncertainty over the value function that relies on inducing a distribution over temporal difference errors. This exploration signal controls for state-action transitions so as to isolate uncertainty in value that is due to uncertainty over the agent's parameters. Because our measure of uncertainty conditions on state-action transitions, we cannot act on this measure directly. Instead, we incorporate it as an intrinsic reward and treat exploration as a separate learning problem, induced by the agent's temporal difference uncertainties. We introduce a distinct exploration policy that learns to collect data with high estimated uncertainty, which gives rise to a "curriculum" that smoothly changes throughout learning and vanishes in the limit of perfect value estimates. We evaluate our method on hard-exploration tasks, including Deep Sea and Atari 2600 environments and find that our proposed form of exploration facilitates efficient exploration.

## 1 INTRODUCTION

Striking the right balance between exploration and exploitation is fundamental to the reinforcement learning problem. A common approach is to derive exploration from the policy being learned. Dithering strategies, such as $\epsilon$-greedy exploration, render a reward-maximising policy stochastic around its reward maximising behaviour (Williams & Peng, 1991). Other methods encourage higher entropy in the policy (Ziebart et al., 2008), introduce an intrinsic reward (Singh et al., 2005), or drive exploration by sampling from the agent's belief over the MDP (Strens, 2000).

While greedy or entropy-maximising policies cannot facilitate temporally extended exploration (Osband et al., 2013; 2016a), the efficacy of intrinsic rewards depends crucially on how they relate to the extrinsic reward that comes from the environment (Burda et al., 2018a). Typically, intrinsic rewards for exploration provide a bonus for visiting novel states (e.g Bellemare et al., 2016) or visiting states where the agent cannot predict future transitions (e.g Pathak et al., 2017; Burda et al., 2018a). Such approaches can facilitate learning an optimal policy, but they can also fail entirely in large environments as they prioritise novelty over rewards (Burda et al., 2018b).

Methods based on the agent's uncertainty over the optimal policy explicitly trade off exploration and exploitation (Kearns & Singh, 2002). Posterior Sampling for Reinforcement Learning (PSRL; Strens, 2000; Osband et al., 2013) is one such approach, which models a distribution over Markov Decision Processes (MDPs). While PSRL is near-optimal in tabular settings (Osband et al., 2013; 2016b), it cannot be easily scaled to complex problems that require function approximators. Prior work has attempted to overcome this by instead directly estimating the agent's uncertainty over the policy's value function (Osband et al., 2016a; Moerland et al., 2017; Osband et al., 2019; O'Donoghue et al., 2018; Janz et al., 2019). While these approaches can scale posterior sampling to complex problems and nonlinear function approximators, estimating uncertainty over value functions introduces issues that can cause a bias in the posterior distribution (Janz et al., 2019).

In response to these challenges, we introduce *Temporal Difference Uncertainties* (TDU), which derives an intrinsic reward from the agent's uncertainty over the value function. Concretely, TDU relies on the Bootstrapped DQN (Osband et al., 2016a) and separates exploration and reward-maximising behaviour into two separate policies that bootstrap from a shared replay buffer. This separation allows us to derive an exploration signal for the exploratory policy from estimates of uncertainty of the reward-maximising policy. Thus, TDU encourages exploration to collect data with high model uncertainty over reward-maximising behaviour, which is made possible by treating exploration as a separate learning problem. In contrast to prior works that directly estimate value function uncertainty, we estimate uncertainty over *temporal difference (TD) errors*. By conditioning on observed state-action transitions, TDU controls for environment uncertainty and provides an exploration signal only insofar as there is model uncertainty. We demonstrate that TDU can facilitate efficient exploration in challenging exploration problems such as Deep Sea and Montezuma's Revenge.

## 2 ESTIMATING VALUE FUNCTION UNCERTAINTY IS HARD

We begin by highlighting that estimating uncertainty over the value function can suffer from bias that is very hard to overcome with typical approaches (see also Janz et al., 2019). Our analysis shows that biased estimates arise because uncertainty estimates require an integration over unknown future state visitations. This requires tremendous model capacity and is in general infeasible. Our results show that we cannot escape a bias in general, but we can take steps to mitigate it by *conditioning* on an observed trajectory. Doing so removes some uncertainty over future state-visitations and we show in Section 3 that it can result in a substantially smaller bias.

We consider a Markov Decision Process $(\mathcal{S}, \mathcal{A}, \mathcal{P}, \mathcal{R}, \gamma)$ for some given state space $(\mathcal{S})$, action space $(\mathcal{A})$, transition dynamics $(\mathcal{P})$, reward function $(\mathcal{R})$ and discount factor $(\gamma)$. For a given (deterministic) policy $\pi : \mathcal{S} \mapsto \mathcal{A}$, the action value function is defined as the expected cumulative reward under the policy starting from state $s$ with action $a$:

$$Q_\pi(s, a) := \mathbb{E}_\pi \left[ \sum_{t=0}^{\infty} \gamma^t r_{t+1} \, \Bigg| \, s_0 = s, a_0 = a \right] = \mathbb{E}_{\substack{r \sim \mathcal{R}(s,a) \\ s' \sim \mathcal{P}(s,a)}} [r + \gamma Q_\pi(s', \pi(s'))], \qquad (1)$$

where $t$ index time and the expectation $\mathbb{E}_\pi$ is with respect to realised rewards $r$ sampled under the policy $\pi$; the right-hand side characterises $Q$ recursively under the Bellman equation. The action-value function $Q_\pi$ is estimated under a function approximator $Q_\theta$ parameterised by $\theta$. Uncertainty over $Q_\pi$ is expressed by placing a distribution over the parameters of the function approximator, $p(\theta)$. We overload notation slightly and write $p(\theta)$ to denote the probability density function $p_\theta$ over a random variable $\theta$. Further, we denote by $\theta \sim p(\theta)$ a random sample $\theta$ from the distribution defined by $p_\theta$. Methods that rely on posterior sampling under function approximators assume that the induced distribution, $p(Q_\theta)$, is an accurate estimate of the agent's uncertainty over its value function, $p(Q_\pi)$, so that sampling $Q_\theta \sim p(Q_\theta)$ is approximately equivalent to sampling from $Q_\pi \sim p(Q_\pi)$.

For this to hold, the moments of $p(Q_\theta)$ at each state-action pair $(s, a)$ must correspond to the expected moments in future states. In particular, moments of $p(Q_\pi)$ must satisfy a Bellman Equation akin to Eq. 1 (O'Donoghue et al., 2018). We focus on the mean $(\mathbb{E})$ and variance $(\mathbb{V})$:

$$\mathbb{E}_\theta[Q_\theta(s, a)] = \mathbb{E}_\theta[\mathbb{E}_{r,s'}[r + \gamma Q_\theta(s', \pi(s'))]], \qquad (2)$$

$$\mathbb{V}_\theta[Q_\theta(s, a)] = \mathbb{V}_\theta[\mathbb{E}_{r,s'}[r + \gamma Q_\theta(s', \pi(s'))]]. \qquad (3)$$

If $\mathbb{E}_\theta[Q_\theta]$ and $\mathbb{V}_\theta[Q_\theta]$ fail to satisfy these conditions, the estimates of $\mathbb{E}[Q_\pi]$ and $\mathbb{V}[Q_\pi]$ are biased, causing a bias in exploration under posterior sampling from $p(Q_\theta)$. Formally, the agent's uncertainty over $p(Q)$ implies uncertainty over the MDP (Strens, 2000). Given a belief over the MDP, i.e., a distribution $p(M)$, we can associate each $M \sim p(M)$ with a distinct value function $Q_\pi^M$. Lemma 1 below shows that, for $p(\theta)$ to be interpreted as representing some $p(M)$ by push-forward to $p(Q_\theta)$, the induced moments must match under the Bellman Equation.

**Lemma 1.** *If $\mathbb{E}_\theta[Q_\theta]$ and $\mathbb{V}_\theta[Q_\theta]$ fail to satisfy Eqs. 2 and 3, respectively, they are biased estimators of $\mathbb{E}_M\left[Q_\pi^M\right]$ and $\mathbb{V}_M\left[Q_\pi^M\right]$ for any choice of $p(M)$.*

All proofs are deferred to Appendix B. Lemma 1 highlights why estimating uncertainty over value functions is so challenging; while the left-hand sides of Eqs. 2 and 3 are stochastic in $\theta$ only, the right-hand sides depend on marginalising over the MDP. This requires the function approximator to generalise to unseen future trajectories. Lemma 1 is therefore a statement about scale; the harder it is to generalise, the more likely we are to observe a bias—even in deterministic environments.

This requirement of "strong generalisation" poses a particular problem for neural networks that tend to interpolate over the training data (e.g. Li et al., 2020; Liu et al., 2020; Belkin et al., 2019), but the issue is more general. In particular, we show that factorising the posterior $p(\theta)$ will typically cause estimation bias for all but tabular MDPs. This is problematic because it is often computationally infeasible to maintain a full posterior; previous work either maintains a full posterior over the final layer of the function approximator (Osband et al., 2016a; O'Donoghue et al., 2018; Janz et al., 2019) or maintains a diagonal posterior over all parameters (Fortunato et al., 2018; Plappert et al., 2018) of the neural network. Either method limits how expressive the function approximator can be with respect to future states, thereby causing an estimation bias. To establish this formally, let $Q_\theta := w \circ \phi_\vartheta$, where $\theta = (w_1, \ldots, w_n, \vartheta_1, \ldots, \vartheta_v)$, with $w \in \mathbb{R}^n$ a linear projection and $\phi : \mathcal{S} \times \mathcal{A} \to \mathbb{R}^n$ a feature extractor with parameters $\vartheta \in \mathbb{R}^v$.

**Proposition 1.** *If the number of state-action pairs where $\mathbb{E}_\theta[Q_\theta(s, a)] \neq \mathbb{E}_\theta[Q_\theta(s', a')]$ is greater than $n$, where $w \in \mathbb{R}^n$, then $\mathbb{E}_\theta[Q_\theta]$ and $\mathbb{V}_\theta[Q_\theta]$ are biased estimators of $\mathbb{E}_M[Q_\pi^M]$ and $\mathbb{V}_M[Q_\pi^M]$ for any choice of $p(M)$.*

This result is a consequence of the feature extractor $\psi$ mapping into a co-domain that is larger than the space spanned by $w$; a bias results from having more unique state-action representations $\psi(s, a)$ than degrees of freedom in $w$. The implication is that function approximators under factorised posteriors cannot generalise uncertainty estimates across states (a similar observation in tabular settings was made by Janz et al., 2019)—they can only produce temporally consistent uncertainty estimates if they have the capacity to memorise point-wise uncertainty estimates for each $(s, a)$, which defeats the purpose of a function approximator. This is a statement about the structure of $p(\theta)$ and holds for any estimation method. Thus, common approaches to uncertainty estimation with neural networks generally fail to provide unbiased uncertainty estimates over the value function in non-trivial MDPs. Proposition 1 shows that to accurately capture value function uncertainty, we need a full posterior over parameters, which is often infeasible. It also underscores that the main issue is the dependence on future state visitation. This motivates Temporal Difference Uncertainties as an estimate of uncertainty *conditioned* on observed state-action transitions.

## 3 TEMPORAL DIFFERENCE UNCERTAINTIES

While Proposition 1 states that we cannot remove this bias unless we are willing to maintain a full posterior $p(\theta)$, we can construct uncertainty estimates that control for uncertainty over future state-action transition. In this paper, we propose to estimate uncertainty over a full transition $\tau := (s, a, r, s')$ to isolate uncertainty due to $p(\theta)$. Fixing a transition, we induce a conditional distribution $p(\delta \mid \tau)$ over Temporal Difference (TD) errors, $\delta(\theta, \tau) := \gamma Q_\theta(s', \pi(s')) + r - Q_\theta(s, a)$, that we characterise by its mean and variance:

$$\mathbb{E}_\delta[\delta \mid \tau] = \mathbb{E}_\theta[\delta(\theta, \tau) \mid \tau] \qquad \text{and} \qquad \mathbb{V}_\delta[\delta \mid \tau] = \mathbb{V}_\theta[\delta(\theta, \tau) \mid \tau]. \qquad (4)$$

Estimators over TD-errors is akin to first-difference estimators of uncertainty over the action-value. They can therefore exhibit smaller bias if that bias is temporally consistent. To illustrate, for simplicity assume that $\mathbb{E}_\theta[Q_\theta]$ consistently over/under-estimates $\mathbb{E}_M[Q_\pi^M]$ by an amount $b \in \mathbb{R}$. The corresponding bias in $\mathbb{E}_\theta[\delta(\theta, \tau) \mid \tau]$ is given by $\text{Bias}(\mathbb{E}_\theta[\delta(\theta, \tau) \mid \tau]) = \text{Bias}(\gamma \mathbb{E}_\theta[Q_\theta(s', \pi(s'))] + r - \mathbb{E}_\theta[Q_\theta(s, a)]) = (\gamma - 1)b$. This bias is close to 0 for typical values of $\gamma$—notably for $\gamma = 1$, $\mathbb{E}_\theta[\delta(\theta, \tau) \mid \tau]$ is unbiased. More generally, unless the bias is constant over time as in the above example, we cannot fully remove the bias when constructing an estimator over a quantity that relies on $Q_\theta$. However, as the above example shows, by conditioning on a state-action transition, we can make it significantly smaller. We formalise this logic in the following result.

**Proposition 2.** *For any $\tau := (s, a, r, s')$ and any $p(M)$, given $p(\theta)$, define the following ratios:*

$$\rho = \text{Bias}\left(\mathbb{E}_\theta[Q_\theta(s', \pi(s'))]\right) / \text{Bias}\left(\mathbb{E}_\theta[Q_\theta(s, a)]\right) \tag{5}$$

$$\phi = \text{Bias}\left(\mathbb{E}_\theta\left[Q_\theta(s', \pi(s'))^2\right]\right) / \text{Bias}\left(\mathbb{E}_\theta\left[Q_\theta(s, a)^2\right]\right) \tag{6}$$

$$\kappa = \text{Bias}\left(\mathbb{E}_\theta[Q_\theta(s', \pi(s'))Q_\theta(s, a)]\right) / \text{Bias}\left(\mathbb{E}_\theta\left[Q_\theta(s, a)^2\right]\right) \tag{7}$$

$$\alpha = \mathbb{E}_M\left[Q_\pi^M(s', \pi(s'))\right] / \mathbb{E}_M\left[Q_\pi^M(s, a)\right]. \tag{8}$$

*If $\rho \in (0, 2/\gamma)$, then $\mathbb{E}_\delta[\delta \mid \tau]$ has lower bias than $\mathbb{E}_\theta[Q_\theta(s, a)]$. Moreover, if $\rho = 1/\gamma$, then $\mathbb{E}_\delta[\delta \mid \tau]$ is unbiased. Additionally, there exists $\rho \approx 1$, $\phi \approx 1$, $\kappa \approx 1$, $\alpha \approx 1$ such that $\mathbb{V}_\theta[\delta(\theta, \tau) \mid \tau]$ have less bias than $\mathbb{V}_\theta[Q_\theta(s, a)]$. In particular, if $\rho = \phi = \kappa = \alpha = 1$, then*

$$|\text{Bias}(\mathbb{V}_\theta[\delta(\theta, \tau) \mid \tau])| = |(\gamma - 1)^2 \text{Bias}(\mathbb{V}_\theta[Q_\theta(s, a)])| < |\text{Bias}(\mathbb{V}_\theta[Q_\theta(s, a)])|. \tag{9}$$

*Further, $\rho = 1/\gamma$, $\kappa = 1/\gamma$, $\phi = 1/\gamma^2$, then $\mathbb{V}_\theta[\delta(\theta, \tau) \mid \tau]$ is unbiased for any $\alpha$.*

The first part of Proposition 2 generalises the example above to cases where the bias $b$ varies across action-state transitions. It is worth noting that the required "smoothness" on the bias is not very stringent: the bias of $\mathbb{E}_\theta[Q_\theta](s', \pi(s'))$ can be twice as large as that of $\mathbb{E}_\theta[Q_\theta](s, a)$ and $\mathbb{E}_\delta[\delta \mid \tau]$ can still produce a less biased estimate. Importantly, it must have the same sign, and so Proposition 2 requires temporal consistency. To establish a similar claim for $\mathbb{V}_\delta[\delta \mid \tau]$, we need a bit more structure. The ratios $\rho$, $\phi$, and $\kappa$ capture temporal consistency in the bias, while $\alpha$ relates to the temporal consistency of the underlying estimand. Proposition 2 establishes that if these ratios are close to unity, then $\mathbb{V}_\theta[\delta(\theta, \tau) \mid \tau]$ will have less bias. For most transitions, it is reasonable to assume that this holds true. In some MDPs, large changes in the reward can cause these requirements to break. Because Proposition 2 only establishes sufficiency, violating this requirement does not necessarily mean that $\mathbb{V}_\delta[\delta \mid \tau]$ has greater bias than $\mathbb{V}_\theta[Q_\theta(s, a)]$. Finally, it is worth noting that these are statements about a given transition $\tau$. In most state-action transitions, the requirements in Proposition 2 will hold, in which case $\mathbb{E}_\delta[\delta \mid \tau]$ and $\mathbb{V}_\delta[\delta \mid \tau]$ exhibit less overall bias. We provide direct empirical support that Proposition 2 holds in practice through careful ceteris paribus comparisons in Section 5.1.

To obtain a concrete signal for exploration, we follow O'Donoghue et al. (2018) and derive an exploration signal from the variance $\mathbb{V}_\theta[\delta(\theta, \tau) | \tau]$. Because $p(\delta \mid \tau)$ is defined per transition, it cannot be used as-is for posterior sampling. Therefore, we incorporate TDU as a signal for exploration via an intrinsic reward. To obtain an exploration signal that is on approximately the same scale as the extrinsic reward, we use the standard deviation $\sigma(\tau) := \sqrt{\mathbb{V}_\theta[\delta(\theta, \tau) \mid \tau]}$ to define an augmented reward function

$$\tilde{\mathcal{R}}(\tau) := \mathcal{R}((s, a) \in \tau) + \beta\,\sigma(\tau), \tag{10}$$

where $\beta \in [0, \infty)$ is a hyper-parameter that determines the emphasis on exploration. Another appealing property of $\sigma$ is that it naturally decays as the agent converges on a solution (as model uncertainty diminishes); TDU defines a distinct MDP $(\mathcal{S}, \mathcal{A}, \mathcal{P}, \tilde{\mathcal{R}}, \gamma)$ under Eq. 10 that converges on the true MDP in the limit of no model uncertainty. For a given policy $\pi$ and distribution $p(Q_\theta)$, there exists an exploration policy $\mu$ that collects transitions over which $p(Q_\theta)$ exhibits maximal uncertainty, as measured by $\sigma$. In hard exploration problems, the exploration policy $\mu$ can behave fundamentally differently from $\pi$. To capture such distinct exploration behaviour, we treat $\mu$ as a separate exploration policy that we train to maximise the augmented reward $\tilde{\mathcal{R}}$, along-side training a policy $\pi$ that maximises the extrinsic reward $\mathcal{R}$. This gives rise to a natural separation of exploitation and exploration in the form of a cooperative multi-agent game, where the exploration policy is tasked with finding experiences where the agent is uncertain of its value estimate for the greedy policy $\pi$. As $\pi$ is trained on this data, we expect uncertainty to vanish (up to noise). As this happens, the exploration policy $\mu$ is incentivised to find new experiences with higher estimated uncertainty. This induces a particular pattern where exploration will reinforce experiences until the agent's uncertainty vanishes, at which point the exploration policy expands its state visitation further. This process can allow TDU to overcome estimation bias in the posterior—since it is in effect exploiting it—in contrast to previous methods that do not maintain a distinct exploration policy. We demonstrate this empirically both on Montezuma's Revenge and on Deep Sea (Osband et al., 2020).

## 4 IMPLEMENTING TDU WITH BOOTSTRAPPING

The distribution over TD-errors that underlies TDU can be estimated using standard techniques for probability density estimation. In this paper, we leverage the statistical bootstrap as it is both easy to implement and provides a robust approximation without requiring distributional assumptions. TDU is easy to implement under the statistical bootstrap—it requires only a few lines of extra code. It can be implemented with value-based as well as actor-critic algorithms (we provide generic pseudo code in Appendix A); in this paper, we focus on $Q$-learning. $Q$-learning alternates between policy evaluation (Eq. 1) and policy improvement under a greedy policy $\pi_\theta(s) = \arg\max_a Q_\theta(s, a)$. Deep $Q$-learning (Mnih et al., 2015) learns $Q_\theta$ by minimising its TD-error by stochastic gradient descent on transitions sampled from a replay buffer. Unless otherwise stated, in practice we adopt a common approach of evaluating the action taken by the learned network through a target network with separate parameters that are updated periodically (Van Hasselt et al., 2016).

Our implementation starts from the bootstrapped DQN (Osband et al., 2016a), which maintains a set of $K$ function approximators $\mathcal{Q} = \{Q_{\theta^k}\}_{k=1}^K$, each parameterised by $\theta^k$ and regressed towards a unique target function using bootstrapped sampling of data from a shared replay memory. The Bootstrapped DQN derives a policy $\pi_\theta$ by sampling $\theta$ uniformly from $\mathcal{Q}$ at the start of each episode. We provide an overview of the Bootstrapped DQN in Algorithm 1 for reference. To implement TDU in this setting, we make a change to the loss function (Algorithm 2, changes highlighted in green). First, we estimate the TDU signal $\sigma$ using bootstrapped value estimation. We estimate $\sigma$ through observed TD-errors $\{\delta_k\}_{k=1}^K$ incurred by the ensemble $\mathcal{Q}$ on a given transition:

$$\sigma(\tau) \approx \sqrt{\frac{1}{K-1} \sum_{k=1}^K \left(\delta(\theta_k, \tau) - \bar{\delta}(\tau)\right)^2},$$

(11)

where $\bar{\delta} = \gamma \bar{Q}' + r - \bar{Q}$, with $\bar{x} := \frac{1}{K}\sum_{i=1}^K x_i$ and $Q' := Q(s', \pi(s'))$. An important assumption underpinning the bootstrapped estimation is that of stochastic optimism (Osband et al., 2016b), which requires the distribution over $\mathcal{Q}$ to be approximately as wide as the true distribution over value estimates. If not, uncertainty over $\mathcal{Q}$ can collapse, which would cause $\sigma$ to also collapse. To prevent this, $\mathcal{Q}$ can be endowed with a prior (Osband et al., 2018) that maintains diversity in the ensemble by defining each value function as $Q_{\theta^k} + \lambda P_k$, $\lambda \in [0, \infty)$, where $P_k$ is a random prior function.

Rather than feeding this exploration signal back into the value functions in $\mathcal{Q}$, which would create a positive feedback loop (uncertainty begets higher reward, which begets higher uncertainty ad-infinitum), we introduce a separate ensemble of exploration value functions $\tilde{\mathcal{Q}} = \{Q_{\tilde{\theta}^k}\}_{k=1}^N$ that we train over the augmented reward (Eqs. 10 and 11). We derive an exploration policy $\mu_{\tilde{\theta}}$ by sampling exploration parameters $\tilde{\theta}$ uniformly from $\tilde{\mathcal{Q}}$, as in the standard bootstrapped DQN.

In summary, our implementation of TDU maintains $K + N$ value functions. The first $K$ defines a standard Bootstrapped DQN. From these, we derive an exploration signal $\sigma$, which we use to train the last $N$ value functions. At the start of each episode, we proceed as in the standard Bootstrapped DQN and randomly sample a parameterisation $\theta$ from $\mathcal{Q} \cup \tilde{\mathcal{Q}}$ that we act under for the duration of the episode. All value functions are trained by bootstrapping from a single shared replay memory (Algorithm 1); see Appendix A for a complete JAX (Bradbury et al., 2018) implementation. Consequently, we execute the (extrinsic) reward-maximising policy $\pi_{\theta \sim \mathcal{Q}}$ with probability $K/(K+N)$ and the exploration policy $\mu_{\tilde{\theta} \sim \tilde{\mathcal{Q}}}$ with probability $N/(K+N)$. While $\pi$ visits states around current reward-maximising behaviour, $\mu$ searches for data with high model uncertainty. While each population $\mathcal{Q}$ and $\tilde{\mathcal{Q}}$ can be seen as performing Bayesian inference, it is not immediately clear that the full agent admits a Bayesian interpretation. We leave this question for future work.

There are several equally valid implementations of TDU (see Appendix A for generic implementations for value-based learning and policy-gradient methods). In our case, it would be equally valid to define only a single exploration policy (i.e. $N = 1$) and specify the probability of sampling this policy. While this can result in faster learning, a potential drawback is that it restricts the exploratory behaviour that $\mu$ can exhibit at any given time. Using a full bootstrapped ensemble for the exploration policy leverages the behavioural diversity of bootstrapping.

**Algorithm 1** Bootstrapped DQN with TDU

**Require:** $M, \mathcal{L}$: MDP to solve, TDU loss
**Require:** $\beta, K, N, \rho$: hyper-parameters
1: Initialise $\mathcal{B}$: replay buffer
2: Initialise $K + N$ value functions, $\mathcal{Q} \cup \tilde{\mathcal{Q}}$
3: **while** not done **do**
4:     Observe $s$ and choose $Q_k \sim \mathcal{Q} \cup \tilde{\mathcal{Q}}$
5:     **while** episode not done **do**
6:         Take action $a = \arg\max_{\hat{a}} Q_k(s, \hat{a})$
7:         Sample mask $m$, $m_i \sim \text{Bin}(n{=}1, p{=}\rho)$
8:         Enqueue transition $(s, a, r, s', m)$ to $\mathcal{B}$
9:         Optimise $\mathcal{L}(\{\theta^k\}_1^K, \{\tilde{\theta}^k\}_1^N, \gamma, \beta, \mathcal{D} \sim \mathcal{B})$
10:     **end while**
11: **end while**

**Algorithm 2** Bootstrapped TD-loss with TDU.

**Require:** $\{\theta^k\}_1^K, \{\tilde{\theta}^k\}_1^N$: parameters
**Require:** $\gamma, \beta, \mathcal{D}$: hyper-parameters, data
1: Initialise $\ell \leftarrow 0$
2: **for** $s, a, r, s', m \in \mathcal{D}$ **do**
3:     $\tau \leftarrow (s, a, r, s', \gamma)$
4:     Compute $\{\delta_i\}_{i=1}^K = \{\delta(\theta^i, \tau)\}_{i=1}^K$
5:     Compute $\sigma$ from $\{\delta_k\}_{k=1}^K$ (Eq. 11)
6:     Update $\tau$ by $r \leftarrow r + \beta\,\sigma$
7:     Compute $\{\tilde{\delta}_j\}_{j=K+1}^N = \{\delta(\tilde{\theta}^j, \tau)\}_{j=K+1}^N$
8:     $\ell \leftarrow \ell + \sum_{i=1}^K m_i \delta_i^2 + \sum_{j=1}^N m_{K+j} \tilde{\delta}_j^2$
9: **end for**
10: **return:** $\ell \,/\, (2(N + K)|\mathcal{D}|)$

## 5 EMPIRICAL EVALUATION

### 5.1 BEHAVIOUR SUITE

Bsuite (Osband et al., 2020) was introduced as a benchmark for characterising core capabilities of RL agents. We focus on a Deep Sea, which is explicitly designed to test for deep exploration. It is a challenging exploration problem where only one out of $2^N$ policies yields any positive reward. Performance is compared on instances of the environment with grid sizes $N \in \{10, 12, \ldots, 50\}$, with an overall "score" that is the percentage of $N$ for which average regret goes to below 0.9 faster than $2^N$. The stochastic version generates a 'bad' transition with probability $1/N$. This is a relatively high degree of uncertainty since the agent cannot recover from a bad transition in an episode.

For all experiments, we use a standard MLP with $Q$-learning, off-policy replay and a separate target network. See Appendix D for details and TDU results on the full suite. We compare TDU on Deep Sea to a battery of exploration methods, broadly divided into methods that facilitate exploration by (a) sampling from a posterior (Bootstrapped DQN, Noisy Nets (Fortunato et al., 2018), Successor Uncertainties (Janz et al., 2019)) or (b) use an intrinsic reward (Random Network Distillation (RND; Burda et al., 2018b), CTS (Bellemare et al., 2016), and Q-Explore (QEX; Simmons-Edler et al., 2019)). We report best scores obtained from a hyper-parameter sweep for each method. Overall, performance varies substantially between methods; only TDU performs (near-)optimally on both the deterministic and stochastic version. Methods that rely on posterior sampling do well on the deterministic version, but suffer a substantial drop in performance on the stochastic version. As the stochastic version serves to increase the complexity of modelling future state visitation, this is clear evidence that these methods suffer from the estimation bias identified in Section 2. We could not make Q-explore and NoisyNets perform well in the default Bsuite setup, while Successor Uncertainties suffers a catastrophic loss of performance on the stochastic version of DeepSea.

Examining TDU, we find that it facilitates exploration while retaining overall performance except on Mountain Car where $\beta > 0$ hurts performance (Appendix D). For Deep Sea (Figure 2), prior functions are instrumental, even for large exploration bonuses ($\beta \gg 0$). However, for a given prior strength, TDU does better than the BDQN ($\beta = 0$). In the stochastic version of Deep Sea, BDQN suffers a significant loss of performance (Figure 2). As this is a ceteris paribus comparison, this performance difference can be directly attributed to an estimation bias in the BDQN that TDU circumvents through its intrinsic reward. That TDU is able to facilitate efficient exploration despite environment stochasticity demonstrates that it can correct for such estimation errors.

Finally, we verify Proposition 2 experimentally. We compare TDU to versions that estimate uncertainty directly over $\mathcal{Q}$ (full analysis in Appendix D.2). We compare TDU to (a) a version where $\sigma$ is defined as standard deviation over $\mathcal{Q}$ and (b) where $\sigma(\mathcal{Q})$ is used as an upper confidence bound in the policy instead of as an intrinsic reward (Figure 2). Neither matches TDU's performance across Bsuite an in particular on Deep Sea. Being ceteris paribus comparisons, this demonstrates that estimating uncertainty over TD-errors provides a stronger signal for exploration, as per Proposition 2.

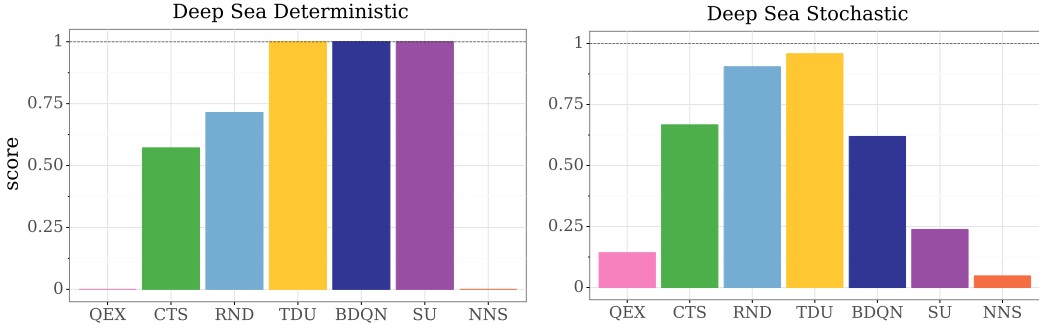

Figure 1: Deep Sea Benchmark. QEX, CTS, and RND use intrinsic rewards; BDQN, SU, and NNS use posterior sampling (Section 5.1). Posterior sampling does well on the deterministic version, but struggles on the stochastic version, suggesting an estimation bias (Section 2). Only TDU performs (near-)optimally on both the deterministic and the stochastic version of Deep Sea.

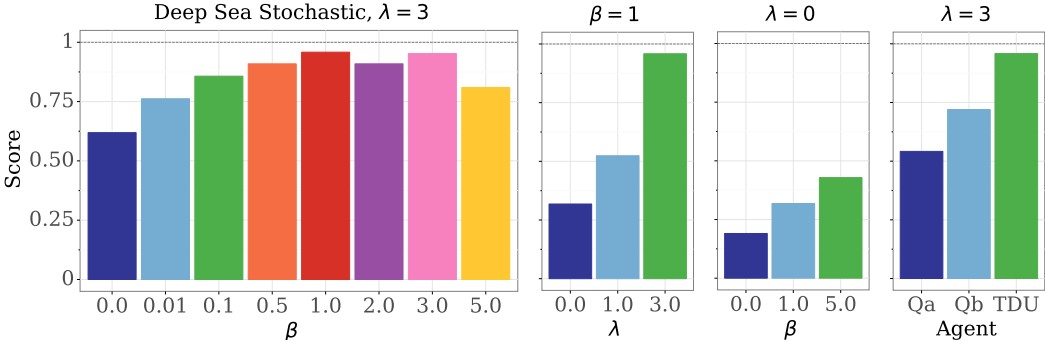

Figure 2: Deep Sea results. All models solve the deterministic version for prior scale $\lambda = 3$ (dashed line). TDU also solves it for $\lambda = 1$. *Left:* introducing stochasticity substantially deteriorates baseline performance; including TDU ($\beta > 0$) recovers close to full performance. *Center left:* effect of varying $\lambda$, TDU benefits from diversity in $Q$ estimates. *Center right*: effect of removing prior ($\lambda = 0$). Increasing $\beta$ improves exploration, but does not reach full performance. *Right:* Qa replaces $\sigma(\delta)$ with $\sigma(\mathcal{Q})$, Qb acts by $\text{argmax}_a(Q + \sigma(\mathcal{Q}))(s, a)$. Estimating uncertainty over $Q$ fails to match TDU.

## 5.2 ATARI

Proposition 1 shows that estimation bias is particularly likely in complex environments that require neural networks to generalise across states. In recent years, such domains have seen significant improvements from running on distributed training platforms that can process large amounts of experience obtained through agent parallelism. It is thus important to develop exploration algorithms that scale gracefully and can leverage the benefits of distributed training. Therefore, we evaluate whether TDU can have a positive impact when combined with the Recurrent Replay Distributed DQN (R2D2) (Kapturowski et al., 2018), which achieves state-of-the-art results on the Atari2600 suite by carefully combining a set of key components: a recurrent state, experience replay, off-policy value learning and distributed training.

As a baseline we implemented a distributed version of the bootstrapped DQN with additive prior functions. We present full implementation details, hyper-parameter choices, and results on all games in Appendix E. For our main results, we run each agent on 8 seeds for 20 billion steps. We focus on games that are well-known to pose challenging exploration problems (Machado et al., 2018): `montezuma_revenge`, `pitfall`, `private_eye`, `solaris`, `venture`, `gravitar`, and `tennis`. Following standard practice, Figure 3 reports Human Normalized Score (HNS), HNS $= \frac{\text{Agent}_{\text{score}} - \text{Random}_{\text{score}}}{\text{Human}_{\text{score}} - \text{Random}_{\text{score}}}$, as an aggregate result across exploration games as well as results on `montezuma_revenge` and `tennis`, which are both known to be particularly hard exploration games (Machado et al., 2018).

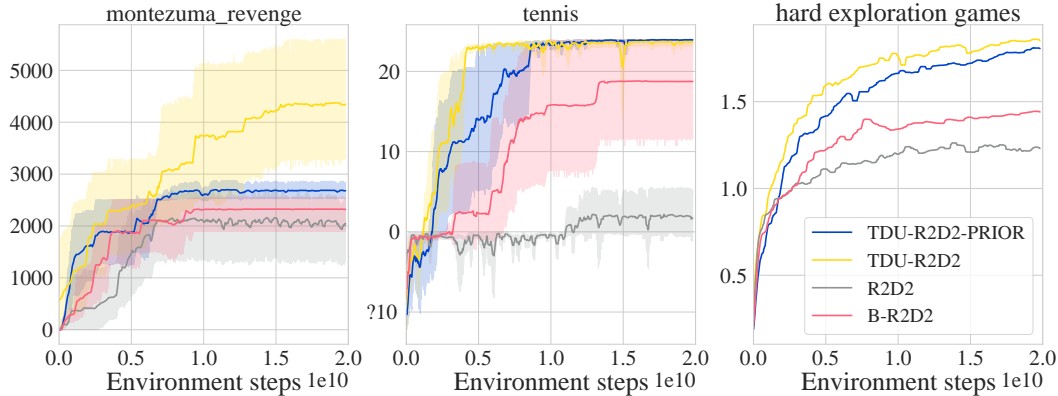

Figure 3: Atari results with distributed training. We compare TDU with and without additive prior functions to R2D2 and Bootstrapped R2D2 (B-R2D2). *Left*: Results for `montezuma_revenge`. *Center*: Results for `tennis`. *Right*: Mean HNS for the hard exploration games in the Atari2600 suite (including `tennis`). Shading depicts standard deviation over 8 seeds.

Generally, we find that TDU facilitates exploration substantially, improving the mean HNS score across exploration games by 30% compared to baselines (right panel, Figure 3). An ANOVA analysis yields a statistically significant difference between TDU and non-TDU methods, controlling for game ($F = 8.17, p = 0.0045$). Notably, TDU achieves significantly higher returns on `montezuma_revenge` and is the only agent that consistently achieves the maximal return on `tennis`. We report all per-game results in Appendix E.4. We observe no significant gains from including prior functions with TDU and find that bootstrapping alone produces relatively marginal gains. Beyond exploration games, TDU can match or improve upon the baseline, but exhibits sensitivity to TDU hyper-parameters ($\beta$, number of explorers ($N$); see Appendix E.3 for details). This finding is in line with observations made by (Puigdomènech Badia et al., 2020); combining TDU with online hyper-parameter adaptation (Schaul et al., 2019; Xu et al., 2018; Zahavy et al., 2020) are exciting avenues for future research. See Appendix E for further comparisons.

In Table 1, we compare TDU to recently proposed state-of-the-art exploration methods. While comparisons must be made with care due to different training regimes, computational budgets, and architectures, we note a general trend that no method is uniformly superior. Methods that are good on extremely sparse exploration games (`montezuma_ revenge` and `pitfall!`) tend to do poorly on games with dense rewards and vice versa. TDU is generally among the top 2 algorithms in all cases except on `montezuma_revenge` and `pitfall!`, state-based exploration is needed to achieve sufficient coverage of the MDP. TDU generally outperforms both Pixel-CNN (Ostrovski et al., 2017), CTS, and RND. TDU is the only algorithm to achieve super-human performance on `solaris` and achieves the highest score of all baselines considered on `venture`.

## 6 RELATED WORK

Bayesian approaches to exploration typically use uncertainty as the mechanism for balancing exploitation and exploration (Strens, 2000). A popular instance of this form of exploration is the PILCO algorithm (Deisenroth & Rasmussen, 2011). While we rely on the bootstrapped DQN (Osband et al., 2016a) in this paper, several other uncertainty estimation techniques have been proposed, such as by placing a parameterised distribution over model parameters (Fortunato et al., 2018; Plappert et al., 2018) or by modeling a distribution over both the value and the returns (Moerland et al., 2017), using Bayesian linear regression on the value function (Azizzadenesheli et al., 2018; Janz et al., 2019), or by modelling the variance over value estimates as a Bellman operation (O'Donoghue et al., 2018). The underlying exploration mechanism in these works is posterior sampling from the agent's current beliefs (Thompson, 1933; Dearden et al., 1998); our work suggests that estimating this posterior is significantly more challenging that previously thought.

An alternative to posterior sampling is to facilitate exploration via learning by introducing an intrinsic reward function. Previous works typically formulate intrinsic rewards in terms of state

Table 1: Atari benchmark on exploration games.[†]Ostrovski et al. (2017), [‡]Bellemare et al. (2016), [◇]Burda et al. (2018b), [⋆]Choi et al. (2018), [§]Puigdomènech Badia et al. (2020), [+]With prior functions.

| Algorithm | Gravitar | Montezuma's Revenge | Pitfall! | Private Eye | Solaris | Venture |
|---|---|---|---|---|---|---|
| Avg. Human | 3,351 | 4,753 | 6,464 | 69,571 | 12,327 | 1,188 |
| R2D2 | 15,680 | 2,061 | 0.0 | 5,322.7 | 3,787.2 | 1,970.7 |
| DQN-PixelCNN[†] | 859.1 | 2,514 | 0.0 | 15,806.5 | 5,501.5 | 1,356.3 |
| DQN-CTS[‡] | 498.3 | 3,706 | 0.0 | 8,358.7 | 82.2 | – |
| RND[◇] | 3,906 | 10,070 | -3 | 8,666 | 3,282 | 1,859 |
| CoEx[⋆] | – | 11,618 | – | 11,000 | – | 1,916 |
| NGU[§] | 14,100 | 10,400 | 8,400 | 100,000 | 4,900 | 1,700 |
| TDU-R2D2 | 13,000 | 5,233 | 0 | 40,544 | 14,712 | 2,000 |
| TDU-R2D2[+] | 10,916 | 2,833 | 0 | 61,168 | 15,230 | 1,977 |

visitation (Lopes et al., 2012; Bellemare et al., 2016; Puigdomènech Badia et al., 2020), state novelty (Schmidhuber, 1991; Oudeyer & Kaplan, 2009; Pathak et al., 2017), or state predictability (Florensa et al., 2017; Burda et al., 2018b; Gregor et al., 2016; Hausman et al., 2018). Most of these works rely on properties of the state space to drive exploration while ignoring rewards. While this can be effective in sparse reward settings (e.g. Burda et al., 2018b; Puigdomènech Badia et al., 2020), it can also lead to arbitrarily bad exploration (see analysis in Osband et al., 2019).

A smaller body of work uses statistics derived from observed rewards (Nachum et al., 2016) or TD-errors to design intrinsic reward functions; our work is particularly related to the latter. Tokic (2010) proposes an extension of $\epsilon$-greedy exploration, where the TD-error modulates $\epsilon$ to be higher in states with higher TD-error. Gehring & Precup (2013) use the mean absolute TD-error, accumulated over time, to measure controllability of a state and reward the agent for visiting states with low mean absolute TD-error. In contrast to our work, this method integrates the TD-error over time to obtain a measure of irreducibility. Simmons-Edler et al. (2019) propose to use two $Q$-networks, where one is trained on data collected under both networks and the other obtains an intrinsic reward equal to the absolute TD-error of the first network on a given transition. In contrast to our work, this method does not have a probabilistic interpretation and thus does not control for uncertainty over the environment. TD-errors have also been used in White et al. (2015), where surprise is defined in terms of the moving average of the TD-error over the full variance of the TD-error. Kumaraswamy et al. (2018) rely on least-squares TD-errors to derive a context-dependent upper-confidence bound for directed exploration. Finally, using the TD-error as an exploration signal is related to the notion of "learnability" or curiosity as a signal for exploration, which is often modelled in terms of the prediction error in a dynamics model (e.g. Schmidhuber, 1991; Oudeyer et al., 2007; Gordon & Ahissar, 2011; Pathak et al., 2017).

## 7 Conclusion

We present Temporal Difference Uncertainties (TDU), a method for estimating uncertainty over an agent's value function. Obtaining well-calibrated uncertainty estimates under function approximation is non-trivial and we show that popular approaches, while in principle valid, can fail to accurately represent uncertainty over the value function because they must represent an unknown future.

This motivates TDU as an estimate of uncertainty conditioned on observed state-action transitions, so that the only source of uncertainty for a given transition is due to uncertainty over the agent's parameters. This gives rise to an intrinsic reward that encodes the agent's model uncertainty, and we capitalise on this signal by introducing a distinct exploration policy. This policy is incentivised to collect data over which the agent has high model uncertainty and we highlight how this separation gives rise to a form of cooperative multi-agent game. We demonstrate empirically that TDU can facilitate efficient exploration in hard exploration games such as Deep Sea and Montezuma's Revenge.

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

## A   IMPLEMENTATION AND CODE

In this section, we provide code for implementing TDU in a general policy-agnostic setting and in the specific case of bootstrapped $Q$-learning. Algorithm 3 presents TDU in a policy-agnostic framework. TDU can be implemented as a pre-processing step (Line 9) that augments the reward with the exploration signal before computing the policy loss. If Algorithm 3 is used to learn a single policy, it benefits from the TDU exploration signal but cannot learn distinct exploration policies for it. In particular, on-policy learning does not admit such a separation. To learn a distinct exploration policy, we can use Algorithm 3 to train the *exploration* policy, while the another policy is trained to maximise extrinsic rewards only using both its own data and data from the exploration policy. In case of multiple policies, we need a mechanism for sampling behavioural policies. In our experiments we settled on uniform sampling; more sophisticated methods can potentially yield better performance.

In the case of value-based learning, TDU takes a special form that can be implemented efficiently as a staggered computation of TD-errors (Algorithm 4). Concretely, we compute an estimate of the distribution of TD-errors from some given distribution over the value function parameters (Algorithm 4, Line 3). These TD-errors are used to compute the TDU signal $\sigma$, which then modulates the reward used to train a $Q$ function (Algorithm 4, Line 7). Because the only quantities being computed are TD-errors, this can be combined into a single error signal (Algorithm 4, Line 11). When implemented under bootstrapping, `Qparams` denotes the ensemble $\mathcal{Q}$ and `Qtilde_distribution_params` denotes the ensemble $\tilde{\mathcal{Q}}$; we compute the loss as in Algorithm 2.

Finally, Algorithm 5 presents a complete JAX (Bradbury et al., 2018) implementation that can be used along with the Bsuite (Osband et al., 2020) codebase.[1] We present the corresponding TDU agent class (Algorithm 4), which is a modified version of the `BootstrappedDqn` class in `bsuite/baselines/jax/boot_dqn/agent.py` and can be used by direct swap-in.

---

**Algorithm 3** Pseudo-code for generic TDU loss

```
1  def loss(transitions, pi_params, Qtilde_distribution_params, beta):
2    # Estimate TD-error distribution.
3    td = array([td_error(p, transitions) for p in sample(Qtilde_distribution_params)])
4
5    # Compute critic loss.
6    td_loss = mean(0.5 * (td ** 2))
7
8    # Compute exploration bonus.
9    transitions.r_t += beta * stop_gradient(std(td, axis=1))
10
11   # Compute policy loss on transition with augmented reward.
12   pi_loss = pi_loss_fn(pi_params, transitions)
13
14   return pi_loss, td_loss
```

---

**Algorithm 4** Pseudo-code for $Q$-learning TDU loss

```
1  def loss(transitions, Q_params, Qtilde_distribution_params, beta):
2    # Estimate TD-error distribution.
3    td_K = array([td_error(p, transitions) for p in sample(Qtilde_distribution_params)])
4
5    # Compute exploration bonus and Q-function reward.
6    transitions.reward_t += beta * stop_gradient(std(td_K, axis=1))
7    td_N = td_error(Q_params, transitions)
8
9    # Combine for overall TD-loss.
10   td_errors = concatenate((td_ex, td_in), axis=1)
11   td_loss = mean(0.5 * (td_errors) ** 2))
12   return td_loss
```

---

[1]Available at: https://github.com/deepmind/bsuite.

**Algorithm 5** JAX implementation of TDU Agent under Bootstrapped DQN

```python
 1  # Copyright 2020 the Temporal Difference Uncertainties as a Signal for Exploration authors.  Licensed under
 2  # the Apache License, Version 2.0 (the "License"); you may not use this file except in compliance with
 3  # the License.  You may obtain a copy of the License at https://www.apache.org/licenses/LICENSE-2.0.
 4  # Unless required by applicable law or agreed to in writing, software distributed under the License is
 5  # distributed on an "AS IS" BASIS, WITHOUT WARRANTIES OR CONDITIONS OF ANY KIND, either express or implied.
 6  # See the License for the specific language governing permissions and limitations under the License.
 7
 8  class TDU(bsuite.baselines.jax.boot_dqn.BootstrappedDqn):
 9
10    def __init__(self, K: int, beta: float, **kwargs: Any):
11      """TDU under Bootstrapped DQN with randomized prior functions."""
12      super(TDU, self).__init__(**kwargs)
13      network, optimizer, N = kwargs['network'], kwargs['optimizer'], kwargs['num_ensemble']
14      noise_scale, discount = kwargs['noise_scale'], kwargs['discount']
15
16      def td(params: hk.Params, target_params: hk.Params,
17             transitions: Sequence[jnp.ndarray]) -> jnp.ndarray:
18        """TD-error with added reward noise + half-in bootstrap."""
19        o_tm1, a_tm1, r_t, d_t, o_t, z_t = transitions
20        q_tm1 = network.apply(params, o_tm1)
21        q_t = network.apply(target_params, o_t)
22        r_t += noise_scale * z_t
23        return jax.vmap(rlax.q_learning)(q_tm1, a_tm1, r_t, discount * d_t, q_t)
24
25      def loss(params: Sequence[hk.Params], target_params: Sequence[hk.Params],
26              transitions: Sequence[jnp.ndarray]) -> jnp.ndarray:
27        """Q-learning loss with TDU."""
28        # Compute TD-errors for first K members.
29        o_tm1, a_tm1, r_t, d_t, o_t, m_t, z_t = transitions
30        td_K = [td(params[k], target_params[k],
31                [o_tm1, a_tm1, r_t, d_t, o_t, z_t[:, k]]) for k in range(K)]
32
33        # TDU signal on first K TD-errors.
34        r_t += beta * jax.lax.stop_gradient(jnp.std(jnp.stack(td_K, axis=0), axis=0))
35
36        # Compute TD-errors on augmented reward for last K members.
37        td_N = [td(params[k], target_params[k],
38                [o_tm1, a_tm1, r_t, d_t, o_t, z_t[:, k]]) for k in range(K, N)]
39
40        return jnp.mean(m_t.T * jnp.stack(td_K + td_N)  ** 2)
41
42      def update(state: TrainingState, gradient: Sequence[jnp.ndarray]) -> TrainingState:
43        """Gradient update on ensemble member."""
44        updates, new_opt_state = optimizer.update(gradient, state.opt_state)
45        new_params = optix.apply_updates(state.params, updates)
46        return TrainingState(params=new_params, target_params=state.target_params,
47                             opt_state=new_opt_state, step=state.step + 1)
48
49      @jax.jit
50      def sgd_step(states: Sequence[TrainingState],
51                   transitions: Sequence[jnp.ndarray]) -> Sequence[TrainingState]:
52        """Does a step of SGD for the whole ensemble over `transitions`."""
53        params, target_params = zip(*[(state.params, state.target_params) for state in states])
54        gradients = jax.grad(loss)(params, target_params, transitions)
55        return [update(state, gradient) for state, gradient in zip(states, gradients)]
56
57      self._sgd_step = sgd_step  # patch BootDQN sgd_step with TDU sgd_step.
58
59    def update(self, timestep: dm_env.TimeStep, action: base.Action,
60               new_timestep: dm_env.TimeStep):
61      """Update the agent: add transition to replay and periodically do SGD."""
62      if new_timestep.last():
63        self._active_head = self._ensemble[np.random.randint(0, self._num_ensemble)]
64
65      mask = np.random.binomial(1, self._mask_prob, self._num_ensemble)
66      noise = np.random.randn(self._num_ensemble)
67      transition = [timestep.observation, action, np.float32(new_timestep.reward),
68                    np.float32(new_timestep.discount), new_timestep.observation, mask, noise]
69      self._replay.add(transition)
70      if self._replay.size < self._min_replay_size:
71        return
72
73      if self._total_steps % self._sgd_period == 0:
74        transitions = self._replay.sample(self._batch_size)
75        self._ensemble = self._sgd_step(self._ensemble, transitions)
76
77      for k, state in enumerate(self._ensemble):
78        if state.step % self._target_update_period == 0:
79          self._ensemble[k] = state._replace(target_params=state.params)
```

# B   PROOFS

We begin with the proof of Lemma 1. First, we show that if Eq. 2 and Eq. 3 fail, $p(\theta)$ induce a distribution $p(Q_\theta)$ whose first two moments are biased estimators of the moments of the distribution of interest $p(Q_\pi)$, for *any* choice of belief over the MDP, $p(M)$. We restate it here for convenience.

**Lemma 1.** *If $\mathbb{E}_\theta[Q_\theta]$ and $\mathbb{V}_\theta[Q_\theta]$ fail to satisfy Eqs. 2 and 3, respectively, they are biased estimators of $\mathbb{E}_M\big[Q_\pi^M\big]$ and $\mathbb{V}_M\big[Q_\pi^M\big]$ for any choice of $p(M)$.*

*Proof.* Assume the contrary, that $\mathbb{E}_M\big[Q_\pi^M(s,\pi(s))\big] = \mathbb{E}_\theta[Q_\theta(s,\pi(s))]$ for all $(s,a) \in \mathcal{S} \times \mathcal{A}$. If Eqs. 2 and 3 do not hold, then for any $\mathbb{M} \in \{\mathbb{E}, \mathbb{V}\}$,

$$\mathbb{M}_M\big[Q_\pi^M(s,\pi(s))\big] = \mathbb{M}_\theta[Q_\theta(s,\pi(s))] \tag{12}$$

$$\neq \mathbb{M}_\theta\left[\mathbb{E}_{\substack{s' \sim \mathcal{P}(s,\pi(s)) \\ r \sim \mathcal{R}(s,\pi(s))}}[r + \gamma Q_\theta(s',\pi(s'))]\right] \tag{13}$$

$$= \mathbb{E}_{\substack{s' \sim \mathcal{P}(s,\pi(s)) \\ r \sim \mathcal{R}(s,\pi(s))}}[r + \gamma \mathbb{M}_\theta[Q_\theta(s',\pi(s'))]] \tag{14}$$

$$= \mathbb{E}_{\substack{s' \sim \mathcal{P}(s,\pi(s)) \\ r \sim \mathcal{R}(s,\pi(s))}}\big[r + \gamma \mathbb{M}_M\big[Q_\pi^M(s',\pi(s'))\big]\big] \tag{15}$$

$$= \mathbb{M}_M\left[\mathbb{E}_{\substack{s' \sim \mathcal{P}(s,\pi(s)) \\ r \sim \mathcal{R}(s,\pi(s))}}\big[r + \gamma Q_\pi^M(s',\pi(s'))\big]\right] \tag{16}$$

$$= \mathbb{M}_M\big[Q_\pi^M(s,\pi(s))\big], \tag{17}$$

a contradiction; conclude that $\mathbb{M}_M\big[Q_\pi^M(s,\pi(s))\big] \neq \mathbb{M}_\theta[Q_\theta(s,\pi(s))]$. Eqs. 13 and 17 use Eqs. 2 and 3; Eqs. 12, 13 and 15 follow by assumption; Eqs. 14 and 16 use linearity of the expectation operator $\mathbb{E}_{r,s'}$ by virtue of $\mathbb{M}$ being defined over $\theta$. As $(s,a,r,s')$ and $p(M)$ are arbitrary, the conclusion follows. ∎

Methods that take inspiration from by PSRL but rely on neural networks typically approximate $p(M)$ by a parameter distribution $p(\theta)$ over the value function. Lemma 1 establishes that the induced distribution $p(Q_\theta)$ under push-forward of $p(\theta)$ must propagate the moments of the distribution $p(Q_\theta)$ consistently over the state-space to be unbiased estimate of $p(Q_\pi^M)$, for any $p(M)$.

With this in mind, we now turn to neural networks and their ability to estimate value function uncertainty in MDPs. To prove our main result, we establish two intermediate results. Recall that we define a function approximator $Q_\theta = w \circ \phi_\vartheta$, where $\theta = (w_1, \ldots, w_n, \vartheta_1, \ldots, \vartheta_v)$; $w \in \mathbb{R}^n$ is a linear layer and $\phi : \mathcal{S} \times \mathcal{A} \to \mathbb{R}^n$ is a feature extractor with parameters $\vartheta \in \mathbb{R}^v$.

As before, let $M$ be an MDP $(\mathcal{S}, \mathcal{A}, \mathcal{P}, \mathcal{R}, \gamma)$ with discrete state and action spaces. We denote by $N$ the number of states and actions with $\mathbb{E}_\theta[Q_\theta(s,a)] \neq \mathbb{E}_\theta[Q_\theta(s',a')]$ with $\mathcal{N} \subset \mathcal{S} \times \mathcal{A} \times \mathcal{S} \times \mathcal{A}$ the set of all such pairs $(s,a,s',a')$. This set can be thought of as a minimal MDP—the set of states within a larger MDP where the function approximator generates unique predictions. It arises in an MDP through dense rewards, stochastic rewards, or irrevocable decisions, such as in Deep Sea. Our first result is concerned with a very common approach, where $\vartheta$ is taken to be a point estimate so that $p(\theta) = p(w)$. This approach is often used for large neural networks, where placing a posterior over the full network would be too costly (Osband et al., 2016a; O'Donoghue et al., 2018; Azizzadenesheli et al., 2018; Janz et al., 2019).

**Lemma 2.** *Let $p(\theta) = p(w)$. If $N > n$, with $w \in \mathbb{R}^n$, then $\mathbb{E}_\theta[Q_\theta]$ fail to satisfy the first moment Bellman equation (Eq. 2). Further, if $N > n^2$, then $\mathbb{V}_\theta[Q_\theta]$ fail to satisfy the second moment Bellman equation (Eq. 3).*

*Proof.* Write the first condition of Eq. 2 as

$$\mathbb{E}_\theta\big[w^T \phi_\vartheta(s,a)\big] = \mathbb{E}_\theta\big[\mathbb{E}_{r,s'}\big[r + \gamma w^T \phi_\vartheta(s',\pi(s'))\big]\big]. \tag{18}$$

Using linearity of the expectation operator along with $p(\theta) = p(w)$, we have

$$\mathbb{E}_w[w]^T \phi_\vartheta(s,a) = \mu(s,a) + \gamma \mathbb{E}_w[w]^T \mathbb{E}_{s'}[\phi_\vartheta(s',\pi(s'))], \tag{19}$$

where $\mu(s,a) = \mathbb{E}_{r\sim\mathcal{R}(s,a)}[r]$. Rearrange to get

$$\mu(s,a) = \mathbb{E}_w[w]^T \left( \phi_\vartheta(s,a) - \gamma \mathbb{E}_{s'}[\phi_\vartheta(s',\pi(s'))] \right). \tag{20}$$

By assumption $\mathbb{E}_\theta[Q_\theta(s,a)] \neq \mathbb{E}_\theta[Q_\theta(s',a')]$, which implies $\phi_\vartheta(s,a) \neq \phi_\vartheta(s',\pi(s'))$ by linearity in $w$. Hence $\phi_\vartheta(s,a) - \gamma \mathbb{E}_{s'}[\phi_\vartheta(s',\pi(s'))]$ is non-zero and unique for each $(s,a)$. Thus, Eq. 20 forms a system of linear equations over $\mathcal{S} \times \mathcal{A}$, which can be reduced to a full-rank system over $\mathcal{N}$: $\mu = \Phi \mathbb{E}_w[w]$, where $\mu \in \mathbb{R}^N$ stacks expected reward $\mu(s,a)$ and $\Phi \in \mathbb{R}^{N \times n}$ stacks vectors $\phi_\vartheta(s,a) - \gamma \mathbb{E}_{s'}[\phi_\vartheta(s',\pi(s'))]$ row-wise. Because $\Phi$ is full rank, if $N > n$, this system has no solution. The conclusion follows for $\mathbb{E}_\theta[Q_\theta]$. If the estimator of the mean is used to estimate the variance, then the estimator of the variance is biased. For an unbiased mean, using linearity in $w$, write the condition of Eq. 3 as

$$\mathbb{E}_\theta\left[ \left[ (w - \mathbb{E}_w[w])^T \phi_\vartheta(s,a) \right]^2 \right] = \mathbb{E}_\theta\left[ \left[ \gamma(w - \mathbb{E}_w[w])^T \mathbb{E}_{s'}[\phi_\vartheta(s',\pi(s'))] \right]^2 \right]. \tag{21}$$

Let $\tilde{w} = (w^T - \mathbb{E}_w[w])$, $x = \tilde{w}^T \phi_\vartheta(s,a)$, $y = \gamma \tilde{w}^T \mathbb{E}_{s'}[\phi_\vartheta(s',a')]$. Rearrange to get

$$\mathbb{E}_\theta[x^2 - y^2] = \mathbb{E}_w[(x-y)(x+y)] = 0. \tag{22}$$

Expanding terms, we find

$$0 = \mathbb{E}_\theta\left[ \left( \tilde{w}^T[\phi_\vartheta(s,a) - \gamma \mathbb{E}_{s'}[\phi_\vartheta(s',a')]] \right) \left( \tilde{w}^T[\phi_\vartheta(s,a) + \gamma \mathbb{E}_{s'}[\phi_\vartheta(s',a')]] \right) \right] \tag{23}$$

$$= \sum_{i=1}^n \sum_{j=1}^n \mathbb{E}_w[\tilde{w}_i \tilde{w}_j] d_i^- d_j^+ = \sum_{i=1}^n \sum_{j=1}^n \mathrm{Cov}\,(w_i, w_j) d_i^- d_j^+. \tag{24}$$

where we define $d^- = \phi_\vartheta(s,a) - \gamma \mathbb{E}_{s'}[\phi_\vartheta(s',a')]$ and $d^+ = \phi_\vartheta(s,a) + \gamma \mathbb{E}_{s'}[\phi_\vartheta(s',a')]$. As before, $d^-$ and $d^+$ are non-zero by assumption of unique $Q$-values. Perform a change of variables $\omega_{\alpha(i,j)} = \mathrm{Cov}(w_i, w_j)$, $\lambda_{\alpha(i,j)} = d_i^- d_j^+$ to write Eq. 24 as $0 = \lambda^T \omega$. Repeating the above process for every state and action we have a system $\mathbf{0} = \Lambda \omega$, where $\mathbf{0} \in \mathbb{R}^N$ and $\Lambda \in \mathbb{R}^{N \times n^2}$ are defined by stacking vectors $\lambda$ row-wise. This is a system of linear equations and if $N > n^2$ no solution exists; thus, the conclusion follows for $\mathbb{V}_\theta[Q_\theta]$, concluding the proof. ∎

Note that if $\mathbb{E}_\theta[Q_\theta]$ is biased and used to construct the estimator $\mathbb{E}_\theta[Q_\theta]$, then this estimator is also biased; hence if $N > n$, $p(\theta)$ induce biased estimators $\mathbb{E}_\theta[Q_\theta]$ and $\mathbb{V}_\theta[Q_\theta]$ of $\mathbb{E}_M[Q_\pi^M]$ and $\mathbb{V}_M[Q_\pi^M]$, respectively.

Lemma 2 can be seen as a statement about linear uncertainty. While the result is not too surprising from this point of view, it is nonetheless a frequently used approach to uncertainty estimation. We may hope then that by placing uncertainty over the feature extractor as well, we can benefit from its nonlinearity to obtain greater representational capacity with respect to uncertainty propagation. Such posteriors come at a price. Placing a full posterior over a neural network is often computationally infeasible, instead a common approach is to use a diagonal posterior, i.e. $\mathrm{Cov}(\theta_i, \theta_j) = 0$ (Fortunato et al., 2018; Plappert et al., 2018). Our next result shows that any posterior of this form suffers from the same limitations as placing a posterior only over the final layer. We establish something stronger: any posterior of the form $p(\theta) = p(w)p(\vartheta)$ suffers from the limitations described in Lemma 2.

**Lemma 3.** *Let $p(\theta) = p(w)p(\vartheta)$; if $N > n$, with $w \in \mathbb{R}^n$, then $\mathbb{E}_\theta[Q_\theta]$ fail to satisfy the first moment Bellman equation (Eq. 2). Further, if $N > n^2$, then $\mathbb{V}_\theta[Q_\theta]$ fail to satisfy the second moment Bellman equation (Eq. 3).*

*Proof.* The proof largely proceeds as in the proof of Lemma 2. Re-write Eq. 19 as

$$\mathbb{E}_w[w]^T \mathbb{E}_\vartheta[\phi_\vartheta(s,a)] = \mu(s,a) + \gamma \mathbb{E}_w[w]^T \mathbb{E}_{s'}[\mathbb{E}_\vartheta[\phi_\vartheta(s',\pi(s'))]]. \tag{25}$$

Perform a change of variables $\tilde{\phi} = \mathbb{E}_\vartheta[\phi_\vartheta]$ to obtain

$$\mu(s,a) = \mathbb{E}_w[w]^T \left( \tilde{\phi}(s,a) - \gamma \mathbb{E}_{s'}\left[ \tilde{\phi}(s',\pi(s')) \right] \right). \tag{26}$$

Because $\mathbb{E}_\theta[Q_\theta(s,a)] \neq \mathbb{E}_\theta[Q_\theta(s',a')]$, by linearity in $w$ we have that $\tilde{\phi}(s,a) - \tilde{\phi}(s',a')$ is non-zero for any $(s',a')$ and hence Eq. 26 has no trivial solutions. Proceeding as in the proof of Lemma 2 obtains $\mu = \tilde{\Phi}\mathbb{E}_w[w]$, where $\tilde{\Phi}$ is analogously defined. Note that if $N > n$ there is no solution $\mathbb{E}_w[w]$ for *any* admissible (full-rank) choice of $\tilde{\Phi}$, and hence the conclusion follows for the first part. For the second part, using that $\mathbb{E}_\theta = \mathbb{E}_w\mathbb{E}_\vartheta$ in Eq. 24 yields

$$0 = \sum_{i=1}^{n}\sum_{j=1}^{n} \mathbb{E}_w[\tilde{w}_i\tilde{w}_j]\, \mathbb{E}_\vartheta\left[d_i^- d_j^+\right] = \sum_{i=1}^{n}\sum_{j=1}^{n} \text{Cov}\,(w_i, w_j)\, \mathbb{E}_\vartheta\left[d_i^- d_j^+\right]. \tag{27}$$

Perform a change of variables $\tilde{\lambda}_{\alpha(i,j)} = \mathbb{E}_\vartheta\left[d_i^- d_j^+\right]$. Again, by $\mathbb{E}_\theta[Q_\theta(s,a)] \neq \mathbb{E}_\theta[Q_\theta(s',a')]$ we have that $\tilde{\lambda}$ is non-zero; proceed as before to complete the proof. ∎

We are now ready to prove our main result. We restate it here for convenience:

**Proposition 1.** *If the number of state-action pairs where $\mathbb{E}_\theta[Q_\theta(s,a)] \neq \mathbb{E}_\theta[Q_\theta(s',a')]$ is greater than $n$, where $w \in \mathbb{R}^n$, then $\mathbb{E}_\theta[Q_\theta]$ and $\mathbb{V}_\theta[Q_\theta]$ are biased estimators of $\mathbb{E}_M\left[Q_\pi^M\right]$ and $\mathbb{V}_M\left[Q_\pi^M\right]$ for any choice of $p(M)$.*

*Proof.* Let $p(\theta)$ be of the form $p(\theta) = p(w)$ or $p(\theta) = p(w)p(\vartheta)$. By Lemmas 2 and 3, $p(\theta)$ fail to satisfy Eq. 2. By Lemma 1, this causes $\mathbb{E}_\theta[Q_\theta]$ to be a biased estimator of $\mathbb{E}_M\left[Q_\pi^M\right]$. This in turn implies that $\mathbb{V}_\theta[Q_\theta]$ is a biased estimator of $\mathbb{V}_M\left[Q_\pi^M\right]$. Further, if $N > n^2$, $\mathbb{V}_\theta[Q_\theta]$ is biased independently of $\mathbb{E}_\theta[Q_\theta]$. ∎

We now turn to analysing the bias of our proposed estimators. As before, we will build up to Proposition 2 through a series of lemmas. For the purpose of these results, let $B : \mathcal{S} \times \mathcal{A} \to \mathbb{R}$ denote the bias of $\mathbb{E}_\theta[Q_\theta]$ in any tuple $(s,a) \in \mathcal{S} \times \mathcal{A}$, so that $\text{Bias}(\mathbb{E}_\theta[Q_\theta]\,(s,a)) = B(s,a)$.

**Lemma 4.** *Given a transition $\tau := (s,a,r,s')$, for any $p(M)$, given $p(\theta)$, if*

$$\frac{B(s',\pi(s'))}{B(s,a)} \in (0, 2/\gamma) \tag{28}$$

*then $\mathbb{E}_\theta[\delta(\theta,\tau) \mid \tau]$ has less bias than $\mathbb{E}_\theta[Q_\theta(s,a)]$.*

*Proof.* From direct manipulation of $\mathbb{E}_\theta[\delta(\theta,\tau) \mid \tau]$, we have

$$\mathbb{E}_\theta[\delta(\theta,\tau) \mid \tau] = \mathbb{E}_\theta[\gamma Q_\theta(s',\pi(s')) + r - Q_\theta(s,a)] \tag{29}$$
$$= \gamma\mathbb{E}_\theta[Q_\theta(s',\pi(s'))] + r - \mathbb{E}_\theta[Q_\theta(s,a)] \tag{30}$$
$$= \gamma\mathbb{E}_M\left[Q_\pi^M(s',\pi(s'))\right] + r - \mathbb{E}_M\left[Q_\pi^M(s,a)\right] + \gamma B(s',\pi(s')) - B(s,a) \tag{31}$$
$$= \mathbb{E}_M\left[\delta_\pi^M(\tau)\right] + \gamma B(s',\pi(s')) - B(s,a). \tag{32}$$

Consequently, $\text{Bias}(\mathbb{E}_\theta[\delta(\theta,\tau) \mid \tau]) = \gamma B(s',\pi(s')) - B(s,a)$ and for this bias to be less than $\text{Bias}(\mathbb{E}_\theta[Q_\theta(s,a)]) = B(s,a)$, we require $|\gamma B(s',\pi(s')) - B(s,a)| < |B(s,a)|$. Let $\rho = B(s',\pi(s'))/B(s,a)$ and write $|(\gamma\rho - 1)B(s,a)| < |B(s,a)|$ from which it follows that for this to hold true, we must have $\rho \in (0, 2/\gamma)$, as to be proved. ∎

We now turn to characterising the conditions under which $\mathbb{V}_\theta[\delta(\theta, \tau) \mid \tau]$ enjoys a smaller bias than $\mathbb{V}_\theta[Q_\theta(s, a)]$. Because the variance term involves squaring the TD-error, we must place some restrictions on the expected behaviour of the $Q$-function to bound the bias. First, as with $B$, let $C : \mathcal{S} \times \mathcal{A} \to \mathbb{R}$ denote the bias of $\mathbb{E}_\theta[Q_\theta^2]$ for any tuple $(s, a) \in \mathcal{S} \times \mathcal{A}$, so that $\mathrm{Bias}(\mathbb{E}_\theta[Q_\theta(s, a)^2]) = C(s, a)$. Similarly, let $D : \mathcal{S} \times \mathcal{A} \times \mathcal{S} \to \mathbb{R}$ denote the bias of $\mathbb{E}_\theta[Q_\theta(s', \pi(s'))Q_\theta(s, a)]$ for any transition $(s, a, s') \in \mathcal{S} \times \mathcal{A} \times \mathcal{S}$.

**Lemma 5.** *For any $\tau$ and any $p(M)$, given $p(\theta)$, define relative bias ratios*

$$\rho = \frac{B(s', \pi(s'))}{B(s, a)}, \quad \phi = \frac{C(s', \pi(s'))}{C(s, a)}, \quad \kappa = \frac{D(s, a, s')}{C(s, a)}, \quad \alpha = \frac{\mathbb{E}_M[Q_\pi^M(s', \pi(s'))]}{\mathbb{E}_M[Q_\pi^M(s, a)]}. \quad (33)$$

*There exists $\rho \approx 1$, $\phi \approx 1$, $\kappa \approx 1$, $\alpha \approx 1$ such that $\mathbb{V}_\theta[\delta(\theta, \tau) \mid \tau]$ have less bias than $\mathbb{V}_\theta[Q_\theta(s, a)]$. In particular, if $\rho = \phi = \kappa = \alpha = 1$, then*

$$|\mathrm{Bias}(\mathbb{V}_\theta[\delta(\theta, \tau) \mid \tau])| = |(\gamma - 1)^2 \mathrm{Bias}(\mathbb{V}_\theta[Q_\theta(s, a)])| < |\mathrm{Bias}(\mathbb{V}_\theta[Q_\theta(s, a)])|. \quad (34)$$

*Further, if $\rho = 1/\gamma$, $\kappa = 1/\gamma$, $\phi = 1/\gamma^2$, then $|\mathrm{Bias}(\mathbb{V}_\theta[\delta(\theta, \tau) \mid \tau])| = 0$ for any $\alpha$.*

*Proof.* We begin by characterising the bias of $\mathbb{V}_\theta[Q_\theta(s, a)]$. Write

$$\mathbb{V}_\theta[Q_\theta(s, a)] = \mathbb{E}_\theta\left[Q(s, a)^2\right] - \mathbb{E}_\theta[Q(s, a)]^2 \quad (35)$$

$$= \mathbb{E}_M\left[Q_\pi^M(s, a)^2\right] + C(s, a) - \left(\mathbb{E}_M\left[Q_\pi^M(s, a)\right] + B(s, a)\right)^2. \quad (36)$$

The squared term expands as

$$\left(\mathbb{E}_M\left[Q_\pi^M(s, a)\right] + B(s, a)\right)^2 = \mathbb{E}_M\left[Q_\pi^M(s, a)\right]^2 + 2\mathbb{E}_M\left[Q_\pi^M(s, a)\right]B(s, a) + B(s, a)^2. \quad (37)$$

Let $A(s, a) = \mathbb{E}_M\left[Q_\pi^M(s, a)\right]B(s, a)$ and write the bias of $\mathbb{V}_\theta[Q_\theta(s, a)]$ as

$$\mathrm{Bias}(\mathbb{V}_\theta[Q_\theta(s, a)]) = C(s, a) + 2A(s, a) + B(s, a)^2. \quad (38)$$

We now turn to $\mathbb{V}_\theta[\delta(\theta, \tau) \mid \tau]$. First note that the reward cancels in this expression:

$$\delta(\theta, \tau) - \mathbb{E}_\theta[\delta(\theta, \tau)] = \gamma Q_\theta(s', \pi(s')) - Q_\theta(s, a) - (\gamma \mathbb{E}_\theta[Q_\theta(s', \pi(s'))] - \mathbb{E}_\theta[Q_\theta(s, a)]). \quad (39)$$

Denote by $x_\theta = \gamma Q_\theta(s', \pi(s')) - Q_\theta(s, a)$ with $\mathbb{E}_\theta[x_\theta] = \gamma \mathbb{E}_\theta[Q_\theta(s', \pi(s'))] - \mathbb{E}_\theta[Q_\theta(s, a)]$. Write

$$\mathbb{V}_\theta[\delta(\theta, \tau) \mid \tau] = \mathbb{E}_\theta\left[(\delta(\theta, \tau) - \mathbb{E}_\theta[\delta(\theta, \tau)])^2\right] \quad (40)$$

$$= \mathbb{E}_\theta\left[(x_\theta - \mathbb{E}_\theta[x_\theta])^2\right] \quad (41)$$

$$= \mathbb{E}_\theta\left[x_\theta^2\right] - \mathbb{E}_\theta[x_\theta]^2 \quad (42)$$

$$= \mathbb{E}_\theta\left[(\gamma Q_\theta(s', \pi(s')) - Q_\theta(s, a))^2\right] - (\gamma \mathbb{E}_\theta[Q_\theta(s', \pi(s'))] - \mathbb{E}_\theta[Q_\theta(s, a)])^2. \quad (43)$$

Eq. 41 uses Eq. 39 and Eq. 43 substitutes back for $x_\theta$. We consider each term in the last expression in turn. For the first term, $\mathbb{E}_\theta\left[(\gamma Q_\theta(s', \pi(s')) - Q_\theta(s, a))^2\right]$, expanding the square yields

$$\gamma^2 \mathbb{E}_\theta\left[Q_\theta(s', \pi(s'))^2\right] - 2\gamma \mathbb{E}_\theta[Q_\theta(s', \pi(s')Q_\theta(s, a)] + \mathbb{E}_\theta\left[Q_\theta(s, a)^2\right]. \quad (44)$$

From this, we obtain the bias as

$$\text{Bias}\left(\mathbb{E}_\theta\left[\left(\gamma Q_\theta(s', \pi(s')) - Q_\theta(s, a)\right)^2\right]\right) = \gamma^2 C(s', \pi(s')) - 2\gamma D(s, a, s') + C(s, a) \quad (45)$$

$$= \left(\gamma^2\phi - 2\gamma\kappa + 1\right)C(s, a). \quad (46)$$

We can compare this term to $C(s, a)$ in the bias of of $\mathbb{V}_\theta[Q_\theta(s, a)]$ (Eq. 38). For the bias term in Eq. 46 to be smaller, we require $\left|\left(\gamma^2\phi - 2\gamma\kappa + 1\right)C(s, a)\right| < |C(s, a)|$ from which it follows that $\left(\gamma^2\phi - 2\gamma\kappa + 1\right) \in (-1, 1)$. In terms of $\phi$, this means

$$\phi \in \left(\frac{2k\gamma - 2}{\gamma^2}, \frac{2k}{\gamma}\right). \quad (47)$$

If the bias term $D$ is close to $C$ ($\kappa \approx 1$), this is approximately the same condition as for $\rho$ in Lemma 4. Generally, as $\kappa$ grows large, $\phi$ must grow small and vice-versa. The gist of this requirement is that the biases should be relatively balanced $\kappa \approx \phi \approx 1$.

For the second term in Eq. 43, recall that $\mathbb{E}_\theta[Q_\theta(s', \pi(s'))] = \mathbb{E}_M\left[Q_\pi^M(s', \pi(s'))\right] + B(s', \pi(s'))$ and $\mathbb{E}_\theta[Q_\theta(s, a)] = \mathbb{E}_M\left[Q_\pi^M(s, a)\right] + B(s, a)$. We have

$$\left(\mathbb{E}_\theta[Q_\theta(s', \pi(s'))] - \mathbb{E}_\theta[Q_\theta(s, a)]\right)^2 = \left((\gamma\alpha - 1)\mathbb{E}_M\left[Q_\pi^M(s, a)\right] + (\gamma\rho - 1)B(s, a)\right)^2, \quad (48)$$

where $\alpha = \mathbb{E}_M\left[Q_\pi^M(s', \pi(s'))\right] / \mathbb{E}_M\left[Q_\pi^M(s, a)\right]$. This expands as

$$(\gamma\alpha - 1)^2\mathbb{E}_M\left[Q_\pi^M(s, a)\right]^2 + 2(\gamma\alpha - 1)(\gamma\rho - 1)\mathbb{E}_M\left[Q_\pi^M(s, a)\right]B(s, a) + (\gamma\rho - 1)^2 B(s, a)^2. \quad (49)$$

Note that from Eq. 34, $\left(\mathbb{E}_M\left[Q_\pi^M(s', \pi(s'))\right] - \mathbb{E}_M\left[Q_\pi^M(s, a)\right]\right)^2 = (\gamma\alpha - 1)^2\mathbb{E}_M\left[Q_\pi^M(s, a)\right]^2$ and so the bias of $\mathbb{V}_\theta[\delta(\theta, \tau) \mid \tau]$ can be written as

$$\text{Bias}(\mathbb{V}_\theta[\delta(\theta, \tau) \mid \tau]) = w_1(\phi, \kappa)C(s, a) + w_2(\alpha, \rho)2A(s, a) + w_3(\rho)B(s, a)^2 \quad (50)$$

where

$$w_1(\phi, \kappa) = \left(\gamma^2\phi - 2\gamma\kappa + 1\right), \quad w_2(\alpha, \rho) = (\gamma\alpha - 1)(\gamma\rho - 1), \quad w_3(\rho) = (\gamma\rho - 1)^2. \quad (51)$$

Note that the bias in Eq. 50 involves the same terms as the bias of $\mathbb{V}_\theta[Q_\theta(s, a)]$ (Eq. 38) but are weighted. Hence, there always exist as set of weights such that $|\text{Bias}(\mathbb{V}_\theta[\delta(\theta, \tau) \mid \tau])| < |\text{Bias}(\mathbb{V}_\theta[Q_\theta(s, a)])|$. In particular, if $\rho = 1/\gamma$, $\kappa = 1/\gamma$, $\phi = 1/\gamma^2$, then $\text{Bias}(\mathbb{V}_\theta[\delta(\theta, \tau) \mid \tau])| = 0$ for any $\alpha$. Further, if $\rho = \alpha = \kappa = \phi = 1$, then we have that $w_1(\phi, \kappa) = w_2(\alpha, \rho) = w_3(\rho) = (\gamma - 1)^2$ and so

$$|\text{Bias}(\mathbb{V}_\theta[\delta(\theta, \tau) \mid \tau])| = |(\gamma - 1)^2\text{Bias}(\mathbb{V}_\theta[Q_\theta(s, a)])| < |\text{Bias}(\mathbb{V}_\theta[Q_\theta(s, a)])|, \quad (52)$$

as desired. ∎

**Proposition 2.** *For any $\tau$ and any $p(M)$, given $p(\theta)$, if $\rho \in (0, 2/\gamma)$, then $\mathbb{E}_\delta[\delta \mid \tau]$ has lower bias than $\mathbb{E}_\theta[Q_\theta(s, a)]$. Additionally, there exists $\rho \approx 1$, $\phi \approx 1$, $\kappa \approx 1$, $\alpha \approx 1$ such that $\mathbb{V}_\theta[\delta(\theta, \tau) \mid \tau]$ have less bias than $\mathbb{V}_\theta[Q_\theta(s, a)]$. In particular, if $\rho = \phi = \kappa = \alpha = 1$, then $|\text{Bias}(\mathbb{V}_\theta[\delta(\theta, \tau) \mid \tau])| = |(\gamma - 1)^2\text{Bias}(\mathbb{V}_\theta[Q_\theta(s, a)])| < |\text{Bias}(\mathbb{V}_\theta[Q_\theta(s, a)])|$. Further, if $\rho = 1/\gamma$, $\kappa = 1/\gamma$, $\phi = 1/\gamma^2$, then $|\text{Bias}(\mathbb{V}_\theta[\delta(\theta, \tau) \mid \tau])| = 0$ for any $\alpha$.*

*Proof.* The first part follows from Lemma 4, the second part follows from Lemma 5. ∎

## C  BINARY TREE MDP

In this section, we make a direct comparison between the Bootstrapped DQN and TDU on the Binary Tree MDP introduced by Janz et al. (2019). In this MDP, the agent has two actions in every state. One action terminates the episode with 0 reward while the other moves the agent one step further up the tree. At the final branch, one leaf yields a reward of 1. Which action terminates the episode and which moves the agent to the next branch is randomly chosen per branch, so that the agent must learn an action map for each branch separately. This is a similar environment to Deep Sea, but simpler in that an episode terminates upon taking a wrong action and the agent does not receive a small negative reward for taking the correct action. We include the Binary Tree MDP experiment to compare the scaling property of TDU as compared to TDU on a well-known benchmark.

We use the default Bsuite implementation[2] of the bootstrapped DQN, with the default architecture and hyper-parameters from the published baseline, reported in Table 2. The agent is composed of a two-layer MLP with RELU activations that approximate $Q(s, a)$ and is trained using experience replay. In the case of the bootstrapped DQN, all ensemble members learn from a shared replay buffer with bootstrapped data sampling, where each member $Q_{\theta^k}$ is a separate MLP (no parameter sharing) that is regressed towards separate target networks. We use Adam (Kingma & Ba, 2015) and update target networks periodically (Table 2).

We run 5 seeds per tree-depth, for depths $L \in \{10, 20, \ldots, 250\}$ and report mean performance in Figure 4. Our results are in line with those of Janz et al. (2019), differences are due to how many gradient steps are taken per episode (our results are between the reported scores for the $1\times$ and $25\times$ versions of the bootstrapped DQN). We observe a clear beneficial effect of including TDU, even for small values of $\beta$. Further, we note that performance is largely monotonically increasing in $\beta$, further demonstrating that the TDU signal is well-behaved and robust to hyper-parameter values.

We study the properties of TDU in Figure 5, which reports performance without prior functions ($\lambda = 0$). We vary $\beta$ and the number of exploration value functions $N$. The total number of value functions is fixed at 20, and so varying $N$ is equivalent to varying the degree of exploration. We note that $N$ has a similar effect to $\beta$, but has a slightly larger tendency to induce over-exploration for large values of $N$.

---

[2]https://github.com/deepmind/bsuite/tree/master/bsuite/baselines/jax/bootdqn.

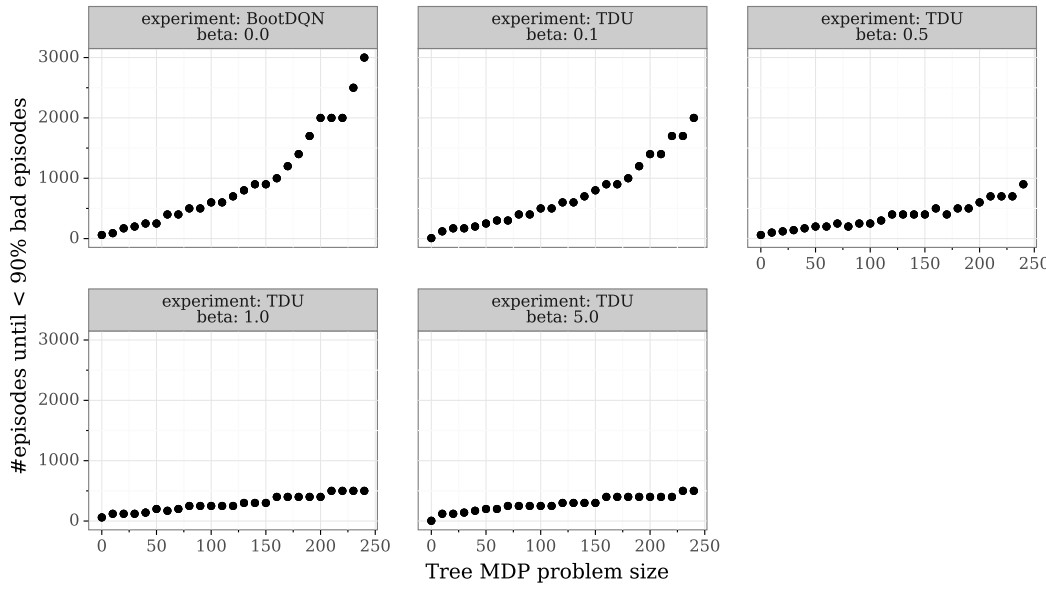

Figure 4: Performance on Binary Tree MDP. *Top left:* BootDQN ($\beta = 0$). *Others:* TDU with varying strenghts of intrinsic reward ($\beta > 0$). Results with prior strength $\lambda = 3$. Mean performance over 5 seeds for each tree depth.

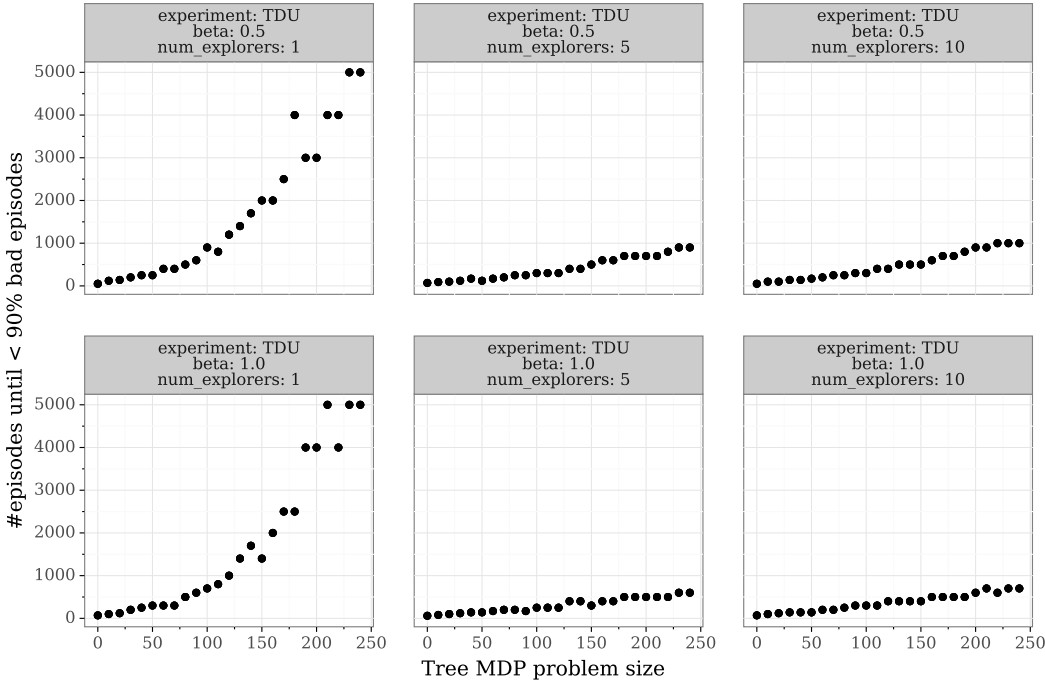

Figure 5: Hyper-parameter sensitivity analysis on Binary Tree MDP. *Top:* Sweep over number of exploration policies ($N$) for $\beta = 0.5$. *Top:* Sweep over number of exploration value functions ($N$) for $\beta = 1$. All results without prior functions ($\lambda = 0$). Mean performance over 5 seeds for each tree depth.

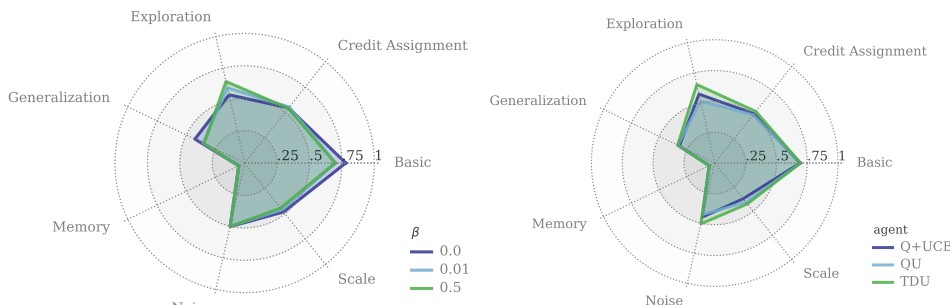

Figure 6: Overall performance scores on Bsuite. *Left:* Effect of varying $\beta$. *Right:* comparison of TDU to exploration under $\sigma = \sigma(\mathcal{Q})$ as intrinsic reward (QU) or as an immediate bonus (Q+UCB).

## D BEHAVIOUR SUITE

From Osband et al. (2020): "The Behaviour Suite for Reinforcement Learning (Bsuite) is a collection of carefully-designed experiments that investigate core capabilities of a reinforcement learning agent . The aim of the Bsuite project is to collect clear, informative and scalable problems that capture key issues in the design of efficient and general learning algorithms and study agent behaviour through their performance on these shared benchmarks."

### D.1 AGENTS AND HYPER-PARAMETERS

All baselines use the default Bsuite DQN implementation[3]. We use the default architecture and hyper-parameters from the published baseline, reported in Table 2, and sweep over algorithm-specific hyper-parameters, reported in Table 3. The agent is composed of a two-layer MLP with RELU activations that approximate $Q(s, a)$ and is trained using experience replay. In the case of the bootstrapped DQN, all ensemble members learn from a shared replay buffer with bootstrapped data sampling, where each member $Q_{\theta^k}$ is a separate MLP (no parameter sharing) that is regressed towards separate target networks. We use Adam (Kingma & Ba, 2015) and update target networks periodically (Table 2).

Table 2: Hyper-parameters for Bsuite.

| | |
|---|---|
| discount factor ($\gamma$) | 0.99 |
| batch size | 32 |
| num hidden layers | 2 |
| hidden layer sizes | [64, 64] |
| ensemble size | 20 |
| learning rate | 0.001 |
| mask prob | 1.0 |
| replay size | 10000 |
| env steps per gradient step | 1 |
| env steps per target update | 4 |

**QEX** Uses two networks $Q_\theta$ and $Q_\vartheta$, where $Q_\theta$ is trained to maximise the extrinsic reward, while $Q_\vartheta$ is trained to maximise the absolute TD-error of $Q_\theta$ (Simmons-Edler et al., 2019). In contrast to TDU, the intrinsic reward is given as a point estimate of the TD-error for a given transition, and thus cannot be interpreted as measuring uncertainty as such.

**CTS** Implements a count-based reward defined by $i(s, a, \mathcal{H}) = (N(s, a, \mathcal{H}) + 0.01)^{-1/2}$, where $\mathcal{H}$ is the history and $N(s, a, \mathcal{H}) = \sum_{\tau \in \mathcal{H}} \mathbf{1}_{(s,a) \in \tau}$ is the number of times $(s, a)$ has appeared in a transition $\tau := (s, a, r, s')$. This intrinsic reward is added to the extrinsic reward to form an augmented reward $\tilde{r} = r + \beta i$ used to train a DQN agent (Bellemare et al., 2016).

**RND** Uses two auxiliary networks $f_\vartheta$ and $f_{\tilde{\vartheta}}$ that map a state into vectors $x = f_\vartheta(s)$ and $\tilde{x} = f_{\tilde{\vartheta}}(s)$, $x, \tilde{x} \in \mathbb{R}^m$. While $\tilde{\vartheta}$ is a random parameter vector that is fixed throughout, $\vartheta$ is trained to minimise the mean squared error $i(s) = \|x - \tilde{x}\|$. This error is simultaneously used as an intrinsic reward in the augmented reward function $\tilde{r}(s, a) = r(s, a) + \beta i(s)$ and is used to train a DQN agent. Following

---

[3]https://github.com/deepmind/bsuite/tree/master/bsuite/baselines/jax/dqn.

Table 3: Hyper-parameter grid searches for Bsuite. Best values in bold.

| Algorithm | Hyper-parameter | Sweep set |
|---|---|---|
| QEX | Intrinsic reward scale ($\beta$) | $\{10^{-4}, 10^{-3}, 10^{-2}, 10^{-1}, 10^0, 5 \cdot 10^0, \mathbf{10^1}, 10^2, 10^3\}$ |
| CTS | Intrinsic reward scale ($\beta$) | $\{10^{-4}, 10^{-3}, 10^{-2}, \mathbf{10^{-1}}, 5 \cdot 10^0, 10^0, 10^1, 10^2, 10^3\}$ |
| RND | Intrinsic reward scale ($\beta$) | $\{10^{-2}, 5 \cdot 10^{-1}, 10^{-1}, 10^0, 5 \times 10^0, \mathbf{10^1}, 10^2\}$ |
| | $x$-dim ($m$) | $\{\mathbf{10}, 64, 128\}$ |
| | Moving average decay ($\alpha$) | $\{0.9, 0.99, \mathbf{0.999}\}$ |
| | Normalise intrinsic reward | $\{\mathbf{True}, False\}$ |
| BDQN | Prior scale ($\lambda$) | $\{0, 1, \mathbf{3}, 5, 10, 50, 100\}$ |
| SU | Hidden size | $\{20, \mathbf{64}\}$ |
| | Likelihood variance ($\beta$) | $\{10^{-2}, 10^{-1}, 10^0, \mathbf{10^1}, 10^2\}$ |
| | Prior variance ($\theta$) | $\{\mathbf{10^{-3}}, 10^{-1}, 10^0, 10^1, 10^3\}$ |
| NNS | Noise scale ($\beta$) | $\{10^{-2}, 10^{-1}, 10^0, \mathbf{10^1}, 10^2\}$ |
| TDU | Prior scale ($\lambda$) | $\{0, 10^0, \mathbf{3 \cdot 10^0}\}$ |
| | Intrinsic reward scale ($\beta$) | $\{10^{-3}, 10^{-2}, 10^{-1}, \mathbf{10^0}, 5 \cdot 10^0, 10^1\}$ |

Burda et al. (2018b), we normalise intrinsic rewards by an exponential moving average of the mean and the standard deviation that are being updated with batch statistics (with decay $\alpha$).

**BDQN** Trains an ensemble $\mathcal{Q} = \{Q_{\theta^k}\}_{k=1}^K$ of DQNs (Osband et al., 2016a). At the start of each episode, one DQN is randomly chosen from which a greedy policy is derived. Data collected is placed in a shared replay memory, and all ensemble members have some probability $\rho$ of training on any transition in the replay. Each ensemble member has its own target network. In addition, each DQN is augmented with a random prior function $f_\vartheta$, where $\tilde{\vartheta}$ is a fixed parameter vector that is randomly sampled at the start of training. Each DQN is defined by $Q_{\theta^k} + \lambda f_{\vartheta^k}$, where $\lambda$ is a hyper-parameter regulating the scale of the prior. Note that the target network uses a distinct prior function.

**SU** Decomposes the DQN as $Q_\theta(s, a) = w^T \psi_\vartheta(s, a)$. The parameters $\vartheta$ are trained to satisfy the Success Feature identity while $w$ is learned using Bayesian linear regression; at the start of each episode, a new $w$ is sampled from the posterior $p(w \mid \text{history})$ (Janz et al., 2019).[4]

**NNS** NoisyNets replace feed-forward layers $Wx + b$ by a noisy equivalent $(W + \Sigma \odot \epsilon^W)x + (b + \sigma \odot \epsilon^b)$, where $\odot$ is element-wise multiplication; $\epsilon_{ij}^W \sim \mathcal{N}(0, \beta)$ and $\epsilon_i^b \sim \mathcal{N}(0, \beta)$ are white noise of the same size as $W$ and $b$, respectively. The set $(W, \Sigma, b, \sigma)$ are learnable parameters that are trained on the normal TD-error, but with the noise vector re-sampled after every optimisation step. Following Fortunato et al. (2018), sample noise separately for the target and the online network.

**TDU** We fix the number of explorers to 10 (half of the number of value functions in the ensemble), which roughly corresponds to randomly sampling between a reward-maximising policy and an exploration policy. Our experiments can be replicated by running the `TDU` agent implemented in Algorithm 5 in the Bsuite GitHub repository.[5]

### D.2 TDU EXPERIMENTS

**Effect of TDU** Our main experiment sweeps over $\beta$ to study the effect of increasing the TDU exploration bonus, with $\beta \in \{0, 0.01, 0.1, 0.5, 1, 2, 3, 5\}$; $\beta = 0$ corresponds to default bootstrapped DQN. We find that $\beta$ reflects the exploitation-exploration trade-off: increasing $\beta$ leads to better performance on exploration tasks (see main paper) but typically leads to worse performance on

---

[4]See `https://github.com/DavidJanz/successor_uncertainties_tabular`.
[5]`https://github.com/deepmind/bsuite/blob/master/bsuite/baselines/jax`.

tasks that do not require further exploration beyond $\epsilon$-greedy (Figure 6). In particular, we find that $\beta > 0$ prevents the agent from learning on Mountain Car, but otherwise retains performance on non-exploration tasks. Figure 7 provides an in-depth comparison per game.

Because $\sigma$ is a principled measure of concentration in the distribution $p(\delta \mid s, a, r, s')$, $\beta$ can be interpreted as specifying how much of the tail of the distribution the agent should care about. The higher we set $\beta$, the greater the agent's sensitivity to the tail-end of its uncertainty estimate. Thus, there is no reason in general to believe that a single $\beta$ should fit all environments, and recent advances in multi-policy learning (Schaul et al., 2019; Zahavy et al., 2020; Puigdomènech Badia et al., 2020) suggests that a promising avenue for further research is to incorporate mechanisms that allow either $\beta$ to dynamically adapt or the sampling probability over policies. To provide concrete evidence to that effect, we conduct an ablation study that uses bandit policy sampling below.

**Effect of prior functions**    We study the inter-relationship between additive prior functions (Osband et al., 2019) and TDU. We sweep over $\lambda \in [0, 1, 3]$, where prior functions define value function estimates by $Q^k = Q_{\theta^k} + \lambda P^k$ for some random network $P^k$. Thus, $\lambda = 0$ implies no prior function. We find a general synergistic relationship; increasing $\lambda$ improves performance (both with and without TDU), and for a given level of $\lambda$, performance on exploration tasks improve for any $\beta > 0$. It should be noted that these effects do no materialise as clearly in our Atari settings, where we find no conclusive evidence to support $\lambda > 0$ under TDU.

**Ablation: exploration under non-TD signals**    To empirically support theoretical underpinnings of TDU (Proposition 2), we conduct an ablation study where $\sigma$ is re-defined as the standard deviation over value estimates:

$$\sigma(\mathcal{Q}) := \sqrt{\frac{1}{K-1} \sum_{k=1}^{K} Q^k - \bar{Q}}. \tag{53}$$

In contrast to TDU, this signal does not condition on the future and consequently is likely to suffer from a greater bias. We apply this signal both as in intrinsic reward (QU), as in TDU, and as an UCB-style exploration bonus (Q+UCB), where $\sigma$ is instead applied while acting by defining a policy by $\pi(\cdot) = \arg\max_a Q(\cdot, a) + \beta\sigma(\mathcal{Q}; \cdot, a)$. Note that TDU cannot be applied in this way because the TDU exploration signal depends on $r$ and $s'$. We tune each baseline over the same set of $\beta$ values as above (incidentally, these coincide to $\beta = 1$) and report best results in Figure 6. We find that either alternative is strictly worse than TDU. They suffer a significant drop in performance on exploration tasks, but are also less able to handle noise and reward scaling. Because the *only* difference between QU and TDU is that in TDU, $\sigma$ conditions on the next state. Thus, there results are in direct support of Proposition 2 and demonstrates that $\mathbb{V}_\theta[\delta \mid \tau]$ is likely to have less bias than $\mathbb{V}_\theta[Q_\theta(s, a)]$.

**Ablation: bandit policy sampling**    Our main results indicate, unsurprisingly, that different environments require different emphasis on exploration. To test this more concretely, in this experiment we replace uniform policy sampling with the UCB1 bandit algorithm. However, in contrast to that example, where UCB1 is used to take actions, here it is used to select a policy for the next episode. We treat each $N + K$ value function as an "arm" and estimate its mean reward $V^k \approx \mathbb{E}_{\pi^k}[r]$, where the expectation is with respect to rewards $r$ collected under policy $\pi^k(\cdot) = \arg\max_a Q^k(\cdot, a)$. The mean reward is estimated as the running average

$$V^k(n) = \frac{1}{n(k)} \sum_{i=1}^{n(k)} r_i, \tag{54}$$

where $n(k)$ is the number of environment steps for which policy $\pi^k$ has been used and $r_i$ are the observed rewards under policy $\pi^k$. Prior to an episode, we choose a policy to act under according to: $\arg\max_{k=1,...,N+K} V^k(n) + \eta\sqrt{\log n/n(k)}$, where $n$ is the total number of environment steps taken so far and $\eta$ is a hyper-parameter that we tune. As in the bandit example, this sampling strategy biases selection towards policies that currently collect higher reward, but balances sampling by a count-based exploration bonus that encourages the agent to eventually try all policies. This bandit mechanism is very simple as our purpose is to test whether some form of adaptive sampling can provide benefits; more sophisticated methods (e.g. Schaul et al., 2019) can yield further gains.

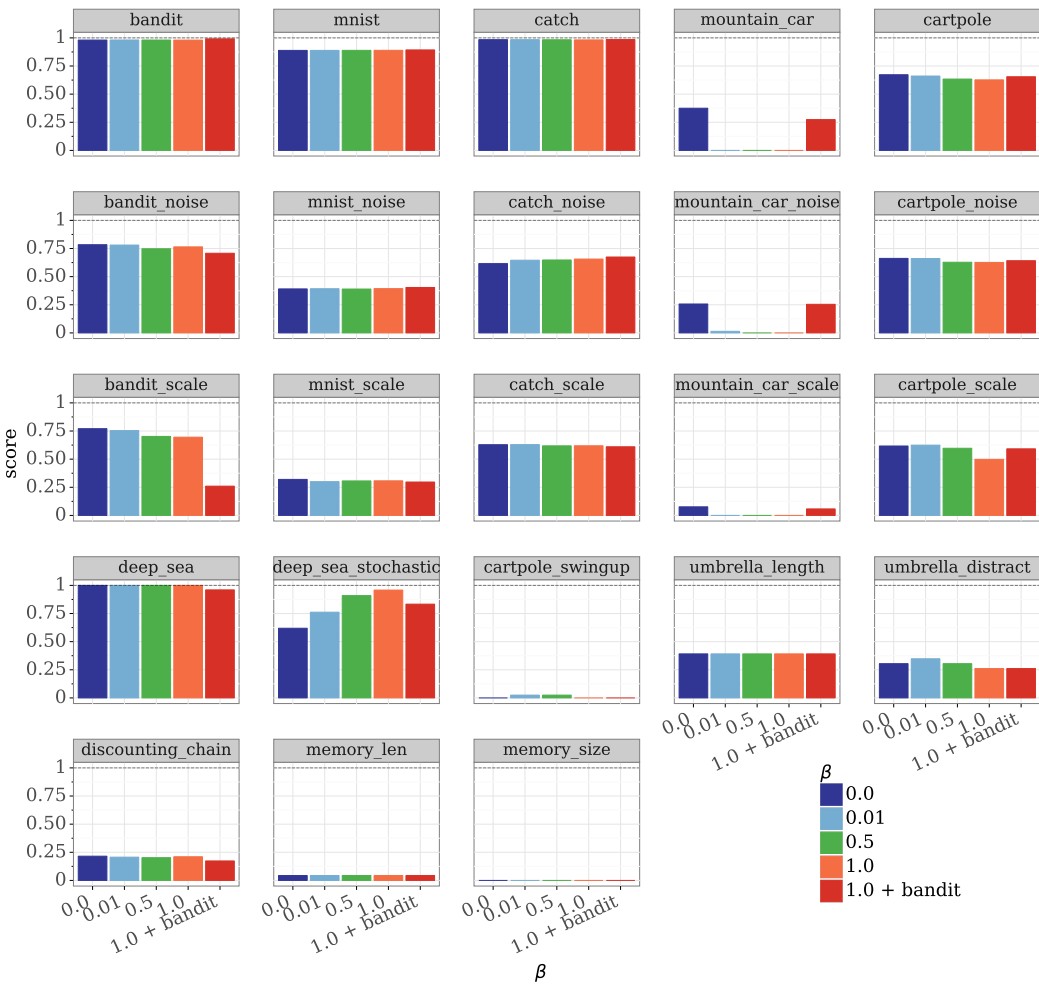

Figure 7: Bsuite per-task results. Results reported for different values of $\beta$ with prior $\lambda = 3$. We also report results under UCB1 policy sampling ("bandit") for $\beta = 1, \lambda = 3, \eta = 8$.

We report full results in Figure 7; we use $\beta = 1$ and tune $\eta \in \{0.1, 1, 2, 4, 6, 8\}$. We report results for the hyper-parameter that performed best overall, $\eta = 8$, though differences with $\eta > 4$ are marginal. While TDU does not impact performance negatively in general, in the one case where it does—Mountain Car—introducing a bandit to adapt exploration can largely recover performance. The bandit yields further gains in dense reward settings, such as in Cartpole and Catch, with an outlying exception in the bandit setting with scaled rewards.

# E   ATARI WITH R2D2

## E.1   BOOTSTRAPPED R2D2

We augment the R2D2 agent with an ensemble of dueling action-value heads $Q_i$. The behavior policy followed by the actors is an $\epsilon$-greedy policy as before, but where the greedy action is determined according to a single $Q_i$ for a fixed length of time (100 actor steps in all of our experiments), before sampling a new $Q_i$ uniformly at random. The evaluation policy is also $\epsilon$-greedy with $\epsilon = 0.001$, where the Q-values are averaged only over the exploiter heads.

Each trajectory inserted into the replay buffer is associated with a binary mask indicating which $Q_i$ will be trained from this data, ensuring that the same mask is used every time the trajectory is sampled. Priorities are computed as in R2D2, except that TD-errors are now averaged over all heads.

Instead of using reward clipping, R2D2 estimates a transformed version of the state-action value function to make it easier to approximate for a neural network. One can define a transformed Bellman operator given any squashing function $h : \mathbb{R} \to \mathbb{R}$ that is monotonically increasing and invertible. We use the function $h : \mathbb{R} \mapsto \mathbb{R}$ defined by

$$h(z) = \text{sign}(z)(\sqrt{|z| + 1} - 1) + \epsilon z, \tag{55}$$

$$h^{-1}(z) = \text{sign}(z) \left( \left( \frac{\sqrt{1 + 4\epsilon(|z| + 1 + \epsilon)} - 1}{2\epsilon} \right) - 1 \right), \tag{56}$$

for $\epsilon$ small. In order to compute the TD errors accurately we need to account for the transformation,

$$\delta(\theta, s, a, r, s') := \gamma h^{-1}(Q_\theta(s', \pi(s'))) + r - h^{-1}(Q_\theta(s, a)). \tag{57}$$

Similarly, at evaluation time we need to apply $h^{-1}$ to the output of each head before averaging.

When making use of a prior we use the form $Q^k = Q_\theta^k + \lambda P^k$, where $P^k$ is of the same architecture as the $Q_\theta^k$ network, but with the widths of all layers cut to reduce computational cost. Finally, instead of n-step returns we utilise $Q(\lambda)$ (Peng & Williams, 1994) as was done in (Guez et al., 2020). In all variants we used the hyper-parameters listed in Table 4.

### E.2 Pre-processing

We used the standard pre-process of the frames received from the Arcade Learning Environment.[6] See Table 5 for details.

### E.3 Hyper-parameter Selection

In the distributed setting we have three TDU-specific hyper-parameters to tune namely: $\beta$, $N$ and the prior weight $\lambda$. For our main results, we run each agent across 8 seeds for 20 billions steps. For ablations and hyper-parameter tuning, we ran agents across 3 seeds for 5 billion environment steps on a subset of 8 games: `frostbite`,`gravitar`, `hero`, `montezuma_revenge`, `ms_pacman`, `seaquest`, `space_invaders`, `venture`. This subset presents quite a bit of diversity including dense-reward games as well as three hard exploration games: `gravitar`,

Table 5: Atari pre-processing hyperparameters.

| | |
|---|---|
| Max episode length | 30 min |
| Num. action repeats | 4 |
| Num. stacked frames | 4 |
| Zero discount on life loss | $false$ |
| Random noops range | 30 |
| Sticky actions | $false$ |
| Frames max pooled | 3 and 4 |
| Grayscaled/RGB | Grayscaled |
| Action set | Full |

`montezuma_revenge` and `venture`. To minimise the computational cost, we started by setting $\lambda$ and $N$ while maintaining $\beta = 1$. We employed a coarse grid of $\lambda \in \{0., 0.05, 0.1\}$ and $N \in \{2, 3, 5\}$. Figure 8 summarises the results in terms of the mean Human Normalised Scores (HNS) across the set. We see that the performance depends on the type of games being evaluated. Specifically, hard exploration games achieve a significantly lower score. Performance does not significantly change with the number of explorers. The largest differences are observed for the exploration games when $N = 5$. We select best performing sets of hyper parameters for TDU with and without additive priors: $(N = 2, \lambda = 0.1)$ and $(N = 5, \lambda = 0)$, respectively.

We evaluate the influence of the exploration bonus strength by fixing $(N = 5, \lambda = 0)$ and choosing $\beta \in \{0.1, 1., 2.\}$. Figure 9 summarises the results. The set of dense rewards is composed of the games in the ablation set that are not considered hard exploration games. We observe that larger values of $\beta$ help on exploration but affect performance on dense reward games. We plot jointly the performance in mean HNS acting when averaging the Q-values for both, the exploiter heads (solid lines) and the explorer heads (dotted lines). We can see that higher strengths for the exploration bonus (higher $\beta$) renders the explorers "uninterested" in the extrinsic rewards, preventing them to converge to exploitative behaviours. This effect is less strong for the hard exploration games.

---

[6]Publicly available at `https://github.com/mgbellemare/Arcade-Learning-Environment`.

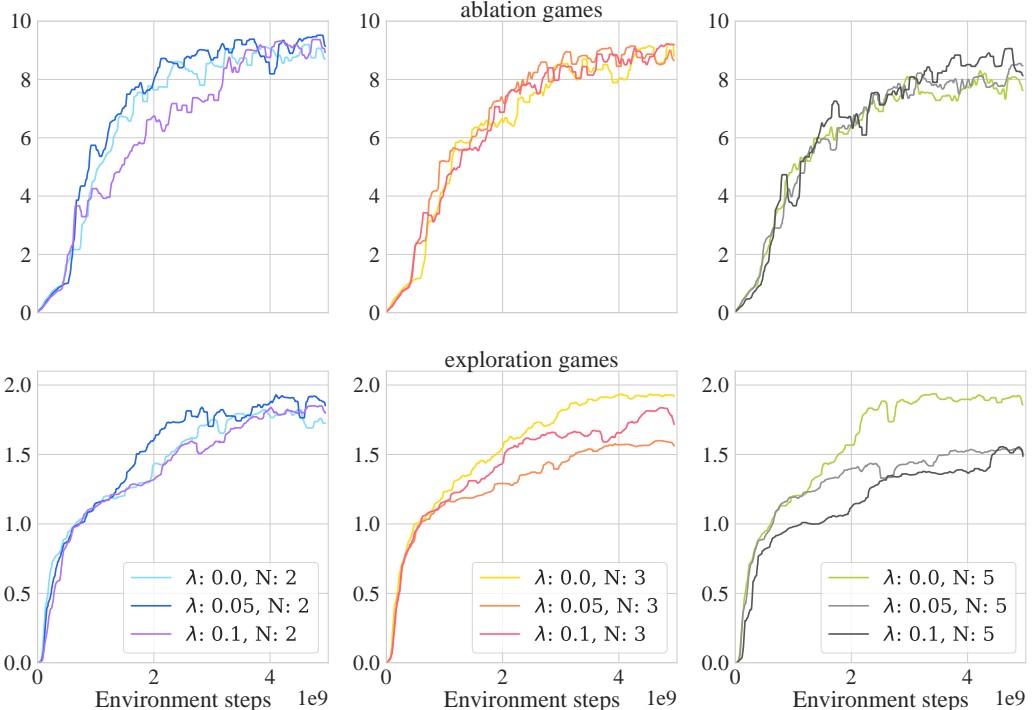

Figure 8: Ablation for prior scale, $\lambda$ and the number of explorers, $N$, on the distributed setting. We fix $\beta = 1$. Refer to the text for details on the ablation and exploration set of games.

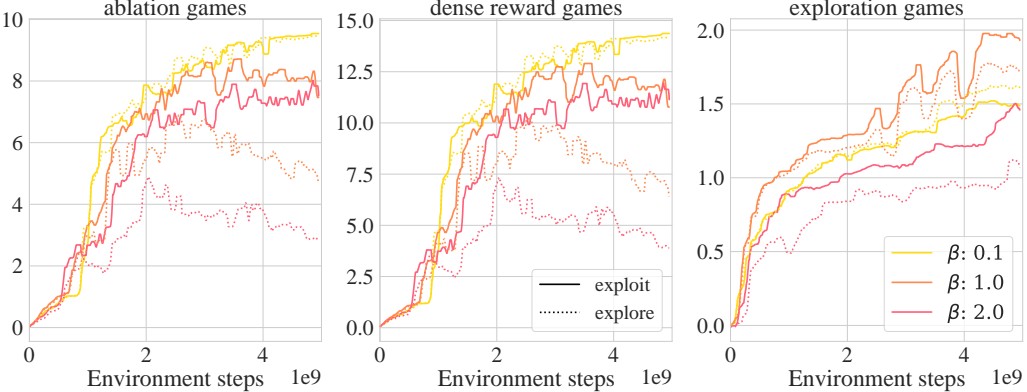

Figure 9: Ablation for the exploration bonus strength, $\beta$, on the distributed setting. We fix ($N = 5, \lambda = 0$). We report the mean HNS for the ensemble of exploiter (solid lines) and the ensemble of explorers (dotted lines). All runs are average over three seeds per game. Refer to the text for details on ablation and exploration set of games.

Figure 10 we show how this effect manifests itself on the performance on three games: `gravitar`, `space_invaders`, and `hero`. This finding also applies to the evaluation performed on our evaluation using all 57 games in the Atari suite, as shown below. We conjecture that controlling for the strength of the exploration bonus on a per game manner would significantly improve the results. This finding is in line with observations made by (Puigdomènech Badia et al., 2020); combining TDU with adaptive policy sampling (Schaul et al., 2019) or online hyper-parameter tuning (Xu et al., 2018; Zahavy et al., 2020) are exciting avenues for future research.

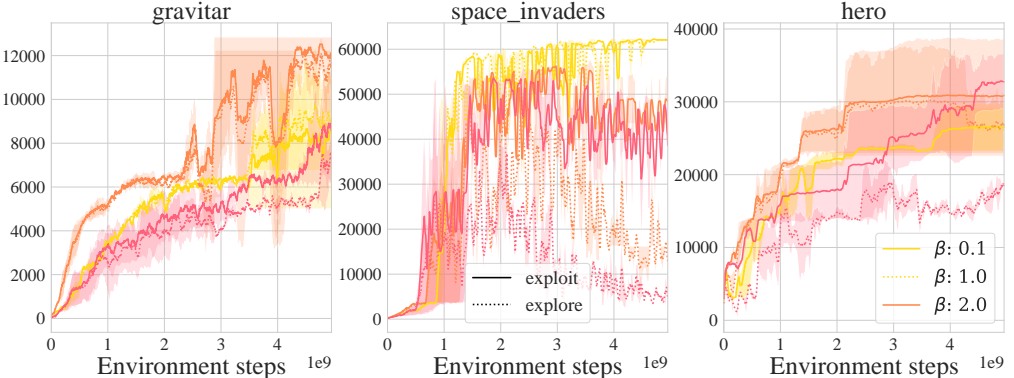

Figure 10: Ablation for the exploration bonus strength, $\beta$, on the distributed setting. We fix ($N = 5, \lambda = 0$). We report the score on three different games for the ensemble of exploiter (solid lines) and the ensemble of explorers (dotted lines). All runs are average over three seeds per game.

### E.4 DETAILED RESULTS: MAIN EXPERIMENT

In this section we provide more detailed results from our main experiment in Section 5.2. We concentrated our attention on the subset of games that are well-known to pose challenging exploration problems (Machado et al., 2018): `montezuma_revenge`, `pitfall`, `private_eye`, `solaris`, `venture`, `gravitar`, and `tennis`. We also add a varied set of dense reward games.

Figure 11 shows the performance for each game. We can see that TDU always performs on par or better than each of the baselines, leading to significant improvements in data efficiency and final score in games such as `montezuma_revenge`, `private_eye`, `venture`, `gravitar`, and `tennis`. Gains in exploration games can be substantial, and in `montezuma_revenge`, `private_eye`, `venture`, and `gravitar`, TDU without prior functions achieves statistically significant improvements. TDU with prior functions achieve statistically significant improvements on `montezuma_revenge`, `private_eye`, and `gravitar`. Beyond this, both methods improve the rate of convergence on `seaquest` and `tennis`, and achieve higher final mean score. Overall, TDU yields benefits across both dense reward and exploration games, as summarised in Figure 12. Note that R2D2's performance on dense reward games is deflated due to particularly low scores on `space_invaders`. Our results are in line with the original publication, where R2D2 does not show substantial improvements until after 35 Bn steps.

### E.5 FULL ATARI SUITE

In this section we report the performance on all 57 games of the Atari suite. In addition to the two configurations used to obtain the results presented in the main text (reported in Section 5.2), in this section we included a variant of each of them with lower exploration bonus strength of $\lambda = 0.1$. In all figures we refer to these variants by adding an L (for lower $\lambda$) at the end of the name, e.g. TDU-R2D2-L. In Figure 13 we report a summary of the results in terms of mean HNS and median HNS for the suite as well as mean HNS restricted to the hard exploration games only. We show the performance on each game in Figure 14. Reducing the value of $\beta$ significantly improves the mean HNS without strongly degrading the performance on the games that are challenging from an exploration standpoint. The difference in performance in terms of mean HNS can be explained by looking at a few high scoring games, for instance: `assault`, `asterix`, `demon_attack` and `gopher` (see Figure 14). We can see that incorporating priors to TDU is not crucial for achieving high performance in the distributed setting.

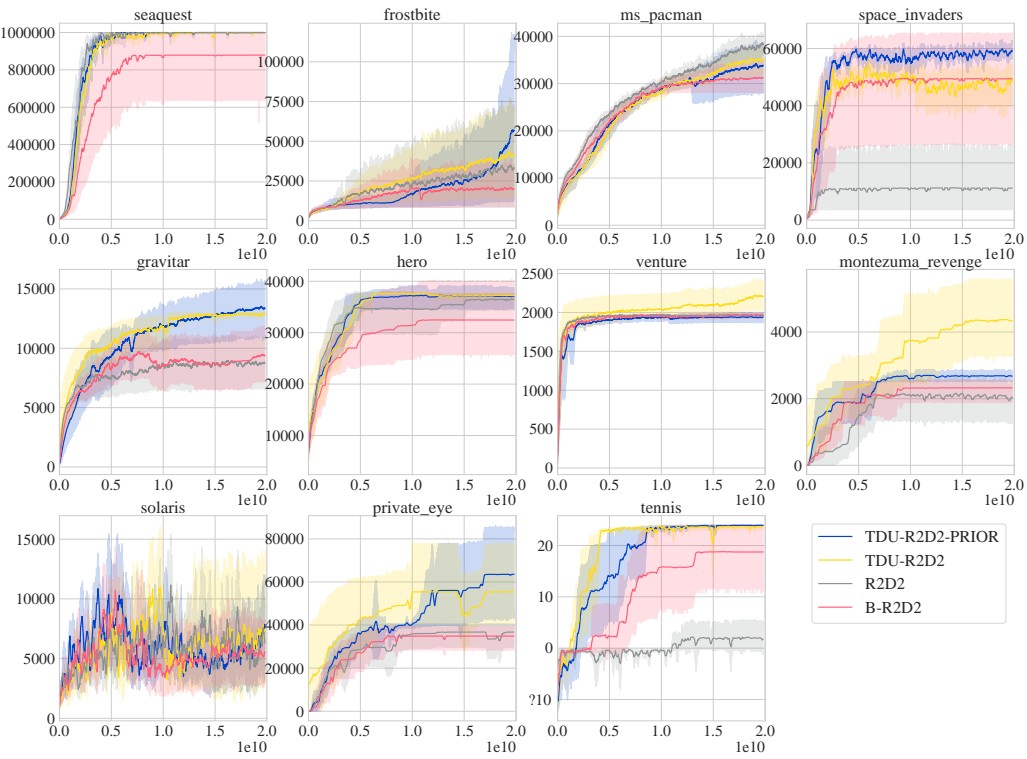

Figure 11: Performance on each game in the main experiment in Section 5.2. Shading depicts standard deviation over 8 seeds.

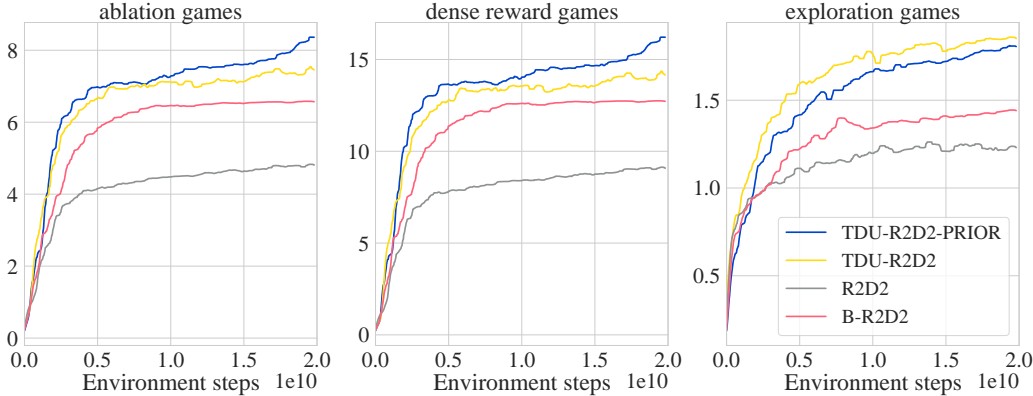

Figure 12: Performance across all games in the main experiment in Section 5.2. We report mean HNS over the full set of games used in the main experiment, dense reward games, and exploration games. Shading depicts standard deviation over 8 seeds.

Table 4: R2D2 hyperparameters.

| | |
|---|---|
| Ensemble size | 10 |
| Optimizer | Adam (Kingma & Ba, 2015) |
| Learning rate | 0.0002 |
| Adam epsilon | 0.001 |
| Adam beta1 | 0.9 |
| Adam beta2 | 0.999 |
| Adam global clip norm | 40 |
| Discount | 0.997 |
| Batch size | 64 |
| Trace length | 80 |
| Replay period | 40 |
| Burn in length | 20 |
| $\lambda$ for RL loss | 0.97 |
| R2D2 reward transformation | $\mathrm{sign}(x) \cdot (\sqrt{|x| + 1} - 1) + 0.001 \cdot x$ |
| Replay capacity (num of sequences) | $1e5$ |
| Replay priority exponent | 0.9 |
| Importance sampling exponent | 0.6 |
| Minimum sequences to start replay | 5000 |
| Actor update period | 100 |
| Target Q-network update period | 400 |
| Evaluation $\epsilon$ | 0.001 |

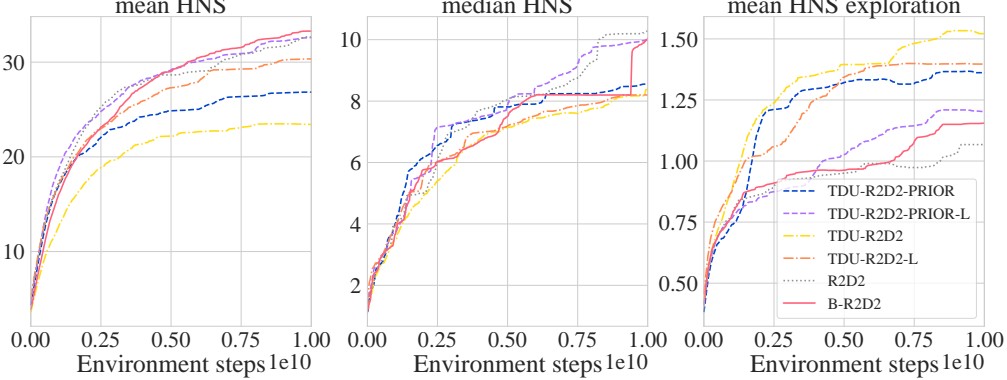

Figure 13: Performance over the 57 atari games. We report mean and median HNS over the full suite, and mean HNS over the exploration games.

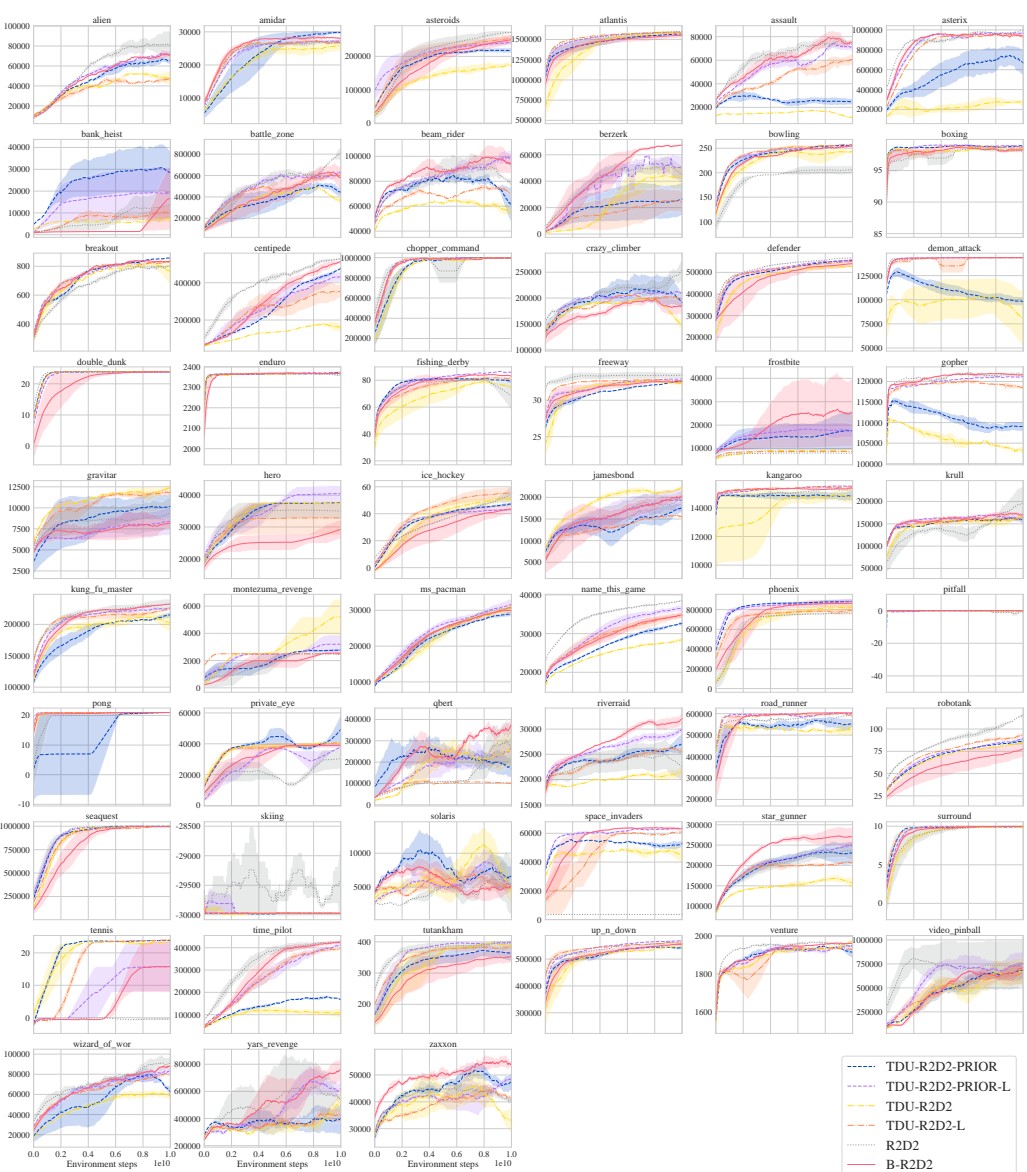

Figure 14: Results for each individual game. Shading depicts standard deviation over 3 seeds.

