# OpenReview forum: "Temporal Difference Uncertainties as a Signal for Exploration"
_ICLR.cc/2021/Conference — Reject_

### Official Review · AnonReviewer4 · 2020-10-20
**Focused and sound**

**Rating:** 5
**Confidence:** 3

**Review:**

Summary:
The paper proposes a novel heuristic approach to estimate the uncertainty of the value function of an agent. Since this heuristic for estimating the uncertainty can be applied not only to tabular RL, but also when using function approximation, it is an interesting approach.

Strong points:
The proposed heuristic can be used to estimate the uncertainty not only for tabular RL, but also when using function approximation, without relying on the ability of the function approximators to specify a value for the uncertainty.

Weak points:
The investigated benchmarks are essentially deterministic and cover a very limited range of the spectrum of RL in general.

Recommendation:
The paper is well written and clearly structured. It introduces a new approach that is likely to be beneficial in a relevant number of problems. I recommend to accept the paper.

Additional feedback with the aim to improve the paper:
„However […] obtaining accurate uncertainty estimates is almost as challenging a problem.“ As challenging as what?
It remains unclear, what is meant by "diverse and deep exploration".
At (\beta >> 0), please use a proper LaTeX command for >>.
Please do correct missing capital letters in the bibliography: e.g.  bayesian, q-


-----------------
(Nov 26.) After reading Review1 I lower my Confidence.
(Nov 29.) Taking into account the other reviews, the authors' responses to these reviews and the discussion, I now think the paper is not quite ready for publication and lower the score to 5.

---

> ### Author Response · Authors · 2020-11-23
> **Thank you for your review.**
>
> Thank you for your review, we are pleased to hear that you find our paper interesting!
>
> In response to your point about experiments, we chose to produce rigorous analyses on Deep Sea and Atari, while also including full results on Bsuite. While Atari is indeed largely deterministic, please do note that the stochastic version of Deep Sea is not. The stochastic version of Deep Sea adds noise to the transition kernel by generating the ‘wrong’ transition with a probability 1 / N. This is a relatively high degree of uncertainty given that if the agent suffers any ‘wrong’ transitions, it cannot get any reward in that episode. Moreover, even if Atari is deterministic, it does pose a significant exploration challenge and the distributed nature of R2D2 injects substantial noise into the learning progress.
>
> Thank you for your additional comments, these have been addressed in the revised manuscript.

---

### Official Review · AnonReviewer3 · 2020-10-28
**Interesting exploration approach**

**Rating:** 7
**Confidence:** 4

**Review:**

Summary:
The authors introduce the use of value function variance (conditioned on state transition) as auxiliary reward promoting exploration during training. The variance is estimated using the bootstrap DQN approach. The main difference with similar methods is that the value uncertainty is not used in a Thompson sampling scheme but it is instead use to provide exploration reward.

Relevance:
The paper address a central problem in deep RL.

Originality:
The central idea is rather original and it elegantly combine ideas the value uncertainty estimation approach with the auxiliary exploration reward strategy. This new form of auxiliary reward is rather attractive as it automatically scales down during training as the model becomes more certain.


Scientific quality:
- The new method is generally well motivated. However, while the author use a Bayesian terminology (e.g. prior, posterior), I am not convinced that the proposed bootstrap method for uncertainty estimation has a clear Bayesian interpretation as a form of approximate inference. Boostrap resampling can lead to proper Bayesian posteriors when the likelihood is used to weight the samples but this does not seem to be the case in this work.
- The experiment section contain an interesting set of experiments and compare with several relevant baselines.
-The paper is clear and very well written. However, I am not convinced by some of the claims. For example, the author claims that since reward uncertainty estimation is biased then networks cannot meaningfully generalize. While the claim can definitely be true, I do not think it follows from their premises. I would like to see the authors to expand this argument.

Pros:
- Simple and elegant new method
- Very good performance when compared with a large number of relevant baselines

Cons:
- Some claims are not completely justified
- It would have been informative to have a wider range of problems in the experiments section

---

> ### Author Response · Authors · 2020-11-23
> **Thank you for your review and support!**
>
> Thank you for your review and feedback; we are glad that you appreciated our paper! Please find answers to your questions below. We use **"lorem ipsum"** to reference parts of your review.
>
> **“I am not convinced that the proposed bootstrap method for uncertainty estimation has a clear Bayesian interpretation as a form of approximate inference.”**
>
> Thank you for pointing this out. We use Bayesian terminology to describe prior works (such as BDQN). We have carefully revised the presentation of TDU to avoid implying that TDU itself is Bayesian. While it relies on uncertainty estimates to prioritise exploration and we do retain BDQNs posterior sampling mechanism, it is not immediately clear that introducing TDU’s intrinsic reward still admits a Bayesian interpretation and establishing this is beyond the scope of our paper.
>
> **“the author claims that since reward uncertainty estimation is biased then networks cannot meaningfully generalize.”**
>
> Our claim is related to Lemma 1 and Proposition 1. Proposition 1 establishes that a bias results from having more unique state-action representations ψ(s,a) than degrees of freedom in the final layer w. From Lemma 1, this means that estimates of uncertainty are biased. In other words, to have unbiased estimates of uncertainty, the final layer w must have as many parameters as there are unique state-action pairs in the MDP, so must be able to produce a tabular representation. By definition, this means that estimators that are subject to Proposition 1 *cannot* generalise uncertainty estimates across state-action pairs, they are only unbiased when they are tabular, which implies no generalisation on the part of the network.
>
> With that said, we do not claim that methods that rely on biased estimates of uncertainty cannot do meaningful exploration. Several of our baselines do perform well on the tasks that we consider, but our experimental results also demonstrates that estimators with smaller bias can yield better exploration.
>
> **"It would have been informative to have a wider range of problems in the experiments section"**
>
> We chose to produce rigorous analyses on Deep Sea as well as the full Bsuite, along with several in-depth analyses of games from Atari and results for all 57 games. It is unclear what types of additional tasks the reviewer is referring to, but we have further added the binary tree MDP in response to reviewer 1.

---

### Official Review · AnonReviewer1 · 2020-10-29

**Rating:** 5
**Confidence:** 4

**Review:**

This paper proposes to use an intrinsic reward based on uncertainties calculated from temporal difference errors. The approach, called Temporal Difference Uncertainties (TDU), estimates the variance of td errors across multiple (bootstrapped) parameters, for a given state, action, next state and reward, where variability is due only to variance in parameters. The other addition is to learn a separate set of action-values that use this intrinsic reward, from the bootstrap set. Actions are then taken by randomly sampling an action-value function from the combined set.

The idea of using td uncertainties as an intrinsic reward is interesting, and should help avoid the fact that bootstrapping value functions alone likely provides insufficient exploration. However, the paper in its current form is not yet ready for publication for two main reasons. First, there are significant gaps in motivating and detailing the TDU technique. In sections 1 and 2, TDU is motivated as an exploration method deriving “an intrinsic reward from the agent’s uncertainty over the value function.”  In particular, it is heavily implied that TDU’s uncertainty estimate does not suffer the bias indicated by Lemma 1.  But it is not detailed how the quantity \sigma(\tau) estimates value function uncertainty, nor how it is unbiased. Why is it a better choice than, for example, directly using the bootstrapped set of action-values? There is some intuition provided, as well as an informal argument following from the distribution p(\delta|\tau) to uncertainty over value function parameters \theta.  But this intuition and lack of formality is at odds with the formality highlighting that estimating value function uncertainty is hard.

Second, the empirical results appear to be obtained with 3 runs and do not provide significant evidence of improvements. Considering the apparent variance in all techniques, no meaningful conclusions may be drawn from comparison between the average of so few samples. The reported shading is also not explained, though I suspect it is standard errors. For only 3 samples, standard errors are not reliable, and in either case here are quite large. It would be better to run on few environments, and try to provide a stronger claim about the role of tdu. Even better would be to also highlight if results change significantly with changes to hyperparameters.

Below, more detailed comments are given about the paper.

It would be useful to better discuss the use of Bootstrapped DQN (BDQN), and why this approach improves on BDQN. In Janz et al. 2019, there is a thoughtful and detailed analysis of one particular failing of BDQN methods (cf. section 5.3) including with a specific demonstrative environment (the binary tree MDP with randomized actions.) While this TDU paper does make a compelling argument that TDU addresses certain challenges of BDQN methods, TDU’s performance on the binary tree MDP (or one sufficiently similar) is not included.  This is a nontrivial omission.  If TDU succeeds on the environment, then a significant challenge is overcome. And if TDU does not succeed, then it is still worthy of publication, just with some discussion.

The theoretical contribution seems like it could be interesting, but it is not fully clear. Is this a statement about all methods that try to estimate some form of value uncertainty? Or those based on PSRL? Lemma 1 is summarized as “the harder it is to generalise, the more likely we are to observe a bias” in value function estimation, which is helpful, but warrants further detail.  A worked, demonstrative example may be edifying here.

It would be useful to discuss more why estimating TDUs is easy (as per the title of Section 3). It looks like the quantity in Equation 4 relies on p(theta), which was stated to be difficult to maintain. A bootstrapped distribution is used to get, as described in Section 4, but Section 3 did not make it clear to me why estimating sigma(tau) was straightforward. There are two additional statements here that could use clarification:
1. Why does using the standard deviation put this bonus on the same scale as the reward?
2. The paper says: “We compare TDU to (a) a version where σ is defined as standard deviation over Q and (b) where σ(Q) is used as an upper confidence bound in the policy, instead of as an intrinsic reward (Figure 2).”  These choices are very interesting, but should be explained in more detail.  Especially why their results indicate that “the key to TDU’s success is that the intrinsic reward relies on a distribution of TD-errors to control for environment uncertainty.“

Minor comments:
1. “neural networks, which are prone to overfitting and tend to generalise through interpolation (e.g. Li et al., 2020; Liu et al., 2020; Belkin et al., 2019)”  This is not an accurate representation of the cited results.  Those papers do not argue that neural networks are prone to overfitting.  Indeed, they argue the opposite: neural networks do not seem to overfit in the overparameterized limit, despite their tendency to interpolate the training set.  That is, generalization is maintained, where high generalization is characterized as per convention: low prediction risk w.r.t. an independently sampled test set.  While this is a significant mischaracterization, it does not appear to be critical to the paper, so we choose not to include it in the rejection justification.

2. The colored bar chart to render parameter sweeps (Fig 2 left, center left, and center right) is a bit distracting. The use of color does not seem to be helping here.

3. The related work section describes and cites some of the same work as the introduction, and is hence some of it is a bit redundant.  There is no need to duplicate that part of the survey in this section.

4. There is an incomplete sentence: “While this can be effective in sparse reward settings, ... it can also lead to arbitrarily bad as the exploration (see analysis in Osband et al., 2019).”

5. In appendix section D.1, there appears to be a missing citation: “Finally, instead of n-step returns we utilize Peng’s Q(λ) as was done in (Value-driven Hindsight Modelling: citation needed).”

6. This work looks at computing variances directly, and using that variance for uncertainty, rather than using the bootstrapped action-values. This citation seems relevant: "Context-Dependent Upper-Confidence Bounds for Directed Exploration", Kumaraswamy et al., 2018.

------- Update
The response and update was a huge improvement. I now much better understand the goals, and the authors added some content that I think made the paper stronger and clearer. One outstanding issue is still that I do not understand why E_M[Q_M^pi] is the gold standard, and why bias is measured relative to that quantity. I explain this more below, but first mention some of the additions I really liked. For an upcoming paper, if this issue is remedied, I think this will be a good paper.

Thank you for introducing Proposition 2 and the explanation about bias beforehand. This very much clarifies the motivation for using TD errors. The result itself highlights that using TDU will have a lower bias to the true expected TD error if the bias for the action-values for (s,a) and (s’,a’) are both in the same direction (both positive or both negative). Otherwise, however, it looks like the bias of TDU is strictly worse. One issue though with this result is that the magnitudes of these quantities could be different. The comparison between bias for TDU and the bias for Q seem a bit like comparing apples and oranges, and I would in fact expect the TD error to be smaller in magnitude and so naturally have a smaller bias. How much is due to this and how much to relative bias reductions? I suspect there is a real bias reduction here, but clarifying this would help.

For the variance result, it seems better to directly report 47, and the discuss ramifications, rather than writing that they all need to be approximately equal. By the way, it is too bad that the result is a bit weaker for the variance, which is precisely the quantity you care about for defining your rewards. But nonetheless this formalization is helpful and provides solid insight.

I like that the empirical work was improved, including adding some comments about significance. I also very much appreciate the ablation, where you use the variance directly from the bootstrapped action-values as an intrinsic reward. My only concern here is that the magnitude of rewards might be quite different, since the TD error should be smaller than the action-values themselves. This might mean different beta are needed.

However, I remain unsure about the importance of this bias that TDU mitigates. You state: “Our analysis shows that biased estimates arise because uncertainty estimates require an integration over unknown future state visitations.“ It remains a strong statement that uncertainty estimates require integration over unknown future states. As one example where this does not seem to be true is the Kumaraswamy paper you have cited. They show that if you use LSTD to get estimates of the action-values for a fixed policy, then you can get an estimate of the variance of the parameters. Maybe this setting assumes too much, and so it does not invalidate your result. But, I do believe a more clear argument is needed in this section for this result, as I expand on below.

“Methods that rely on posterior sampling under function approximators assume that the induced distribution, p(Qθ ), is an accurate estimate of the agent’s uncertainty over its value function, p(Qπ ), so that sampling Qθ ∼ p(Qθ ) is approximately equivalent to sampling from Qπ ∼ p(Qπ ). ” This is a strong statement. I do not see why it is true. Are you suggesting that the agent must have the true Qpi? Is it not enough to use uncertainty estimates (epistemic uncertainty) for a function approximator? This result seems to show: if we want to mimic uncertainty estimates over true action-values for different models, then this is not possible under function approximation. But, that is maybe reasonable. Instead, shouldn’t we ask: how can we mimic uncertainty over approximate action-values for different models? (i.e., relative to our function class)

Additionally, you call E_theta[Q_theta] an estimator and discuss it’s bias. But, isn’t that quantity not random? I presume E_theta[Q_theta] is the expectation, and the difference to E_M[Q_pi] is the bias. But, then what is the estimator that is biased?

Minor comments:

“However, in non-tabular settings that involve function approximators, obtaining accurate uncertainty estimates is almost as challenging as the exploration problem itself. “ What does this mean?
“state-action transitions ” what is that? I think you mean just transition, since you condition on the whole thing
In the proof of Lemma 1, the Variance would remove the expectation over r term. Your result still holds, but the proof itself looks like it should be separated into the two cases.
“While Proposition 1 states that we cannot remove this bias unless we are willing to maintain a full posterior p(θ), ” It is not clear how this result shows this. What if I maintained a full Gaussian posterior over theta? Would that solve the problem? What is a partial posterior?

---

> ### Author Response · Authors · 2020-11-23
> **Thank you for your review; we have made significant revisions to address all your concerns.**
>
> Thank you for your detailed review and feedback. We appreciate that you found our proposed method interesting! We have made major revisions to fully address all your concerns, please see below for full details. We use **"lorem ipsum"**  to reference parts of your review.
>
> **“it is heavily implied that TDU’s uncertainty estimate does not suffer the bias indicated by Lemma 1”**
>
> It is not our intention to imply that TDU solves the bias that arises in Lemma 1. Proposition 1 states that the only solution to this bias is to use a full posterior p(\theta). As this is computationally infeasible for all but small MPDs, our proposal is to estimate uncertainty over a different quantity that mitigates this bias. Here, we use TD-errors. As TD-errors rely on Q_\theta, the same bias arises in TDU (indeed any uncertainty estimate that involves Q_\theta will be subject to the identified in Lemma 1), but because it conditions on a full state-transition it can limit the extent of the bias. In response to your review, we have added a formal analysis to show the conditions under which TDU is an improvement. In brief, \sigma(\tau) has a smaller bias if the bias in E[Q_\theta] is temporally consistent, for instance by persistent overestimation of the value function.
>
> **“it is not detailed how the quantity \sigma(\tau) estimates value function uncertainty, nor how it is unbiased”**
>
> \sigma(\tau) estimates uncertainty over TD-errors, which can be seen as a first-difference estimator of uncertainty over Q. As such, it expresses uncertainty over the value of a policy, but it is not a direct measure of uncertainty over Q_\theta (but rather \Delta Q_\theta. We have updated our wording to clarify this point and avoid further confusion. We did not claim that \sigma is unbiased, but understand that the text might have implied such. In response to your review, we have added a formal analysis to detail how TDU is impacted by a bias in the estimator E[Q] / V[Q].
>
> **“Why is it a better choice than, for example, directly using the bootstrapped set of action-values?”**
>
> The motivation behind TDU is to provide a clearer signal for exploration by better capturing uncertainty in the value function that is due to uncertainty over the agent’s parameters. Our new formal analysis details the scenarios under which this can happen, and shows that, if the bias is temporally consistent, then TDU will have a smaller bias than uncertainty directly over action-values. This is empirically demonstrated on Deep Sea. Because the *only* difference between BDQN and TDU is \sigma, the difference in performance is directly attributable to \sigma. Hence, that BDQN fails to solve the stochastic version of Deep Sea while TDU performs almost as well as in the deterministic case empirically verifies that \sigma provides a signal for exploration that goes beyond sampling from a bootstrapped set of action values. This is then further validated on Atari, where TDU performs significantly better than BDQN on hard exploration games such as Montezuma’s Revenge.
>
> **“3 runs and do not provide significant evidence of improvements [...The reported shading is also not explained]”**
>
> We would like to point out that this refers only to Atari. We run the full Bsuite setup as designed by its authors. In particular, for Deep Sea, we report results from over 100 experiments.
>
> For Atari, we have updated results for a subset of games, including hard exploration games with results from 8 seeds. We also ran these for 20 billion steps to highlight that BDQN saturates while TDU does not. Shading depicts standard deviation, we apologise for this omission, which has been corrected. Our results are statistically significant, as calculated by an ANOVA analysis comparing TDU vs non-TDU methods (BDQN and R2D2), controlling for Atari level (8 seeds per level, 11 levels, F = 8.17, p = 0.0045). We have added these statistics to the revised version of our paper.
>
> **“Even better would be to also highlight if results change significantly with changes to hyperparameters.”**
>
> We provide an ablation of all TDU-specific hyper-parameters for all experiments. On Deep Sea, we have ablations in the main text, Figure 2. We provide further ablations in Appendix, section C.2. Moreover, for Atari, we have a full section on hyper-parameter sensitivity in section D.3, with ablations in Figure 6 and 7.

---

> > ### Author Response · Authors · 2020-11-23
> > **Rebuttal (continued)**
> >
> > **“TDU’s performance on the binary tree MDP (or one sufficiently similar) is not included. This is a nontrivial omission.”**
> >
> > The binary tree MDP is a simplified version of Deep Sea where the episode terminates immediately after the agent takes a sub-optimal action. This makes exploration much easier, because the agent cannot explore further once an incorrect action has been taken. Because we conducted a rigorous analysis on Deep Sea, to satisfy space constraints we did not include this experiment. However, to appease reviewers, we have included results for the binary tree MDP in appendix. TDU improves the scaling factor significantly compared to BDQN. Moreover, TDU removes the need for prior functions in the bootstrap on this domain. Our results demonstrate conclusively that TDU enjoys a significantly superior scaling factor over BDQN.
> >
> > **“The theoretical contribution seems like it could be interesting, but it is not fully clear. Is this a statement about all methods that try to estimate some form of value uncertainty?”**
> >
> > Yes, this is a statement about uncertainty estimation (it does not say anything about how they might be used for exploration, it just so happens that PSRL methods rely heavily on uncertainty over the action-value function). Lemma 1 states that unbiased estimates of uncertainty over the action-value function requires Eqs 1. and 2. to hold. Proposition 1 states that, unless p(\theta) is a full probability distribution (full covariance matrix), then Eqs 1. and 2. will not hold unless the final linear layer has more parameters than there are state-action pairs. This is a statement about the structure of p(\theta) and holds for any method of estimation.
> >
> > **“It would be useful to discuss more why estimating TDUs is easy”**
> >
> > In response to this feedback, we have changed this section title to avoid confusion or to inflate claims. Uncertainty estimation over TD-errors is likely to have a smaller bias than uncertainty estimation over the action-value function, which is the main motivation for our proposed metric.
> >
> > **"Why does using the standard deviation put this bonus on the same scale as the reward?"**
> >
> > The statement in the text is that it is “approximately on the same scale”. This follows because TD-error are approximately on the same scale as the reward: If Q(s, a) has not yet seen a given reward, then the TD-error is approximately equal to the reward, hence on the same order of magnitude as the reward. We take the standard deviation because it is known to be on the same unit of measure as its variable (which is not true for the variance), which for the TD-error means approximately on the same scale as the reward.
> >
> > **“We compare TDU to (a) a version where σ is defined as standard deviation over Q and (b) where σ(Q) is used as an upper confidence bound in the policy, instead of as an intrinsic reward (Figure 2).” These choices are very interesting, but should be explained in more detail. “**
> >
> > We apologise for failing to reference the Appendix, C.2, where we have a substantial analysis that explains these experiments in detail. We have added a reference to the manuscript.
> >
> > **“neural networks, which are prone to overfitting and tend to generalise through interpolation (e.g. Li et al., 2020; Liu et al., 2020; Belkin et al., 2019). This is not an accurate representation of the cited results."**
> >
> > Thank you for this comment. These references are with respect to that neural networks tend to generalise through interpolation, but we understand that it can be read in another way. Our point here refers to strong generalization, as in out-of-distribution generalization, which is extensively discussed in Li et. al. (2020). In particular while they show that neural networks can strongly generalize, this property is by no means a given. We have rephrased the reference to only refer to their tendency to interpolate over training data.
> >
> > Finally, thank you for pointing out the missing references and minor clarification issues. We have rectified these in the revised manuscript.

---

### Official Review · AnonReviewer2 · 2020-10-31
**Interesting work on exploration in RL, needs a clarification on transition function**

**Rating:** 5
**Confidence:** 2

**Review:**

This paper proposes a method for exploration in reinforcement learning by using the uncertainty over the value function as an intrinsic reward. It also offers an interesting theoretical analysis on the problem of estimating uncertainty over the value function. The paper is clearly written in general and has mentioned the related methods sufficiently. Moreover, the authors have compared their method with state-of-the-art models. In the experimental results section, the authors have shown that their model works well both in deterministic and stochastic environment. The noise in this stochastic environment, however, is very low (To be more precise the entropy of transition function is low). This is actually my main question/concern about the paper. In the beginning of section 3, the authors have mentioned that they "fix" the transition function in calculation of uncertainty over the value function. Does this mean that in that part, p(s'|a, s) is set to 1 for the most probable state and 0 for others? If this is the case, the method might not work well when the entropy of the transition function is high (e.g. the agent goes to one state 50% of the time, and to another 50% with an action), or in continuous state space environments where the actions are noisy. In the mentioned cases, sampling cases might actually work better because at least they consider those states in the future. I think it will improve the paper a lot if the authors add such environments to their analysis and discuss such problems.

Update: Thanks the authors for their response. Based on the other reviews and authors' response I decrease my score by 1 point.

---

> ### Author Response · Authors · 2020-11-23
> **Thank you for your review.**
>
> Thank you for your review and feedback, we’re glad you liked our paper!
>
> Thank you for your question. By ‘fix’, we mean that we measure uncertainty over each (s, a, r, s’). In contrast, prior methods estimate uncertainty over each (s, a). In both cases, the estimated uncertainty is obtained by training a (set of) neural network(s) by sampling mini-batches from a replay memory.
>
> To illustrate the difference, consider an example where the agent is taking an action a in a state s and the transition kernel produces a new state s’ with some probability otherwise a new state s’’. In prior methods, the estimated uncertainty would be over the action-value of the point (s, a), so would integrate over the two outcomes. If there is no uncertainty over the outcome s’ and high uncertainty over the outcome s’’, these methods would not be able to differentiate between the two. In contrast, TDU estimates two separate quantities, the uncertainty over the transition (s, a, r’, s’) and over  (s, a, r’’, s’’). Consequently, if there is no uncertainty over (s, a, r’, s’), TDU would be able to make that distinction and produce the appropriate exploration signal. This is important for credit assignment: as we are using action-value functions, the exploration bonus should be attributed to the transition over which the agent is uncertain, rather than a state from which some transitions may induce uncertainty.
>
> As for environment stochasticity, please do note that the stochastic version of Deep Sea adds noise to the transition kernel by generating the ‘wrong’ transition with a probability 1 / N, precisely as in the above example. For Deep Sea, this is a relatively high degree of uncertainty given that if the agent suffers any ‘wrong’ transitions, it cannot get any reward in that episode. We have added a discussion of this point in the revised version of the paper.

---

### Decision · Program_Chairs · 2021-01-07
**Final Decision**

**Decision:**

Reject

**Comment:**

The submitted paper contains interesting theoretical insights into common approaches for exploration and proposes a new way for deriving intrinsic rewards for exploration which is evaluated in several benchmark environments. While all reviewers appreciate these aspects, there are concerns about whether the paper is ready for publication. In particular, the authors’ response did not clarify all open questions and concerns (although the authors already improved the paper a lot by updating the submitted paper according to recommendations/questions of the reviewers). After discussions and author feedback, 3 knowledgable reviewers suggest (weak) rejection of the paper and 1 reviewer suggested acceptance of the paper. Considering this, I recommend to reject the paper but I would like to encourage the authors to consider the comments of the reviewers to revise their paper accordingly, as I expect the paper to then turn into a strong and impactful one.